# DEFINING AND EXTRACTING GENERALIZABLE INTERACTION PRIMITIVES FROM DNNS

**Lu Chen**[1*]  **Siyu Lou**[1,2*]  **Benhao Huang**[1]  **Quanshi Zhang**[1†]
[1]Shanghai Jiao Tong University, Shanghai, China [2]Eastern Institute of Technology, Ningbo, China
{lu.chen,siyu.lou,hbh001098hbh,zqs1022}@sjtu.edu.cn

## ABSTRACT

Faithfully summarizing the knowledge encoded by a deep neural network (DNN) into a few symbolic primitive patterns without losing much information represents a core challenge in explainable AI. To this end, Ren et al. (2024) have derived a series of theorems to prove that the inference score of a DNN can be explained as a small set of interactions between input variables. However, the lack of generalization power makes it still hard to consider such interactions as faithful primitive patterns encoded by the DNN. Therefore, given different DNNs trained for the same task, we develop a new method to extract interactions that are shared by these DNNs. Experiments show that the extracted interactions can better reflect common knowledge shared by different DNNs[1].

## 1 INTRODUCTION

Explaining and quantifying the *exact knowledge* encoded by a deep neural network (DNN) presents a new challenge in explainable AI. Previous studies mainly visualized patterns encoded by DNNs (Bau et al., 2017; Kim et al., 2018) and estimated a saliency map on input variables (Simonyan et al., 2013; R. Selvaraju et al., 2017). However, a new question is that *can we formulate the implicit knowledge encoded by the DNN* as *explicit and symbolic primitive patterns?* In fact, we hope these primitive patterns serve as elementary units for inference, just like *concepts in human cognition*.

However, there is no widely accepted way to define the concept encoded by a DNN, because we cannot mathematically define/formulate the exact concept in human cognition. Nevertheless, if we ignore cognitive issues, Ren et al. (2024); Li & Zhang (2023b) have derived a series of theorems as convincing evidence to take interactions as symbolic primitives encoded by a DNN. Specifically, an interaction captures the intricate nonlinear relationship encoded by the DNN. For instance, when a DNN processes a sentence "*It is raining cats and dogs!*", the DNN may encode the interaction between a set of input variables $S = \{raining, cats, and, dogs\} \subseteq N$. When all words in $S$ are present, an interactive effect $I(S)$ emerges, and pushes the DNN's inference towards the semantic meaning of "heavy rain." However, if any word in $S$ is masked, the effect will be removed.

Ren et al. (2024) have mainly proven two theorems to justify the convincingness of considering above interactions as primitive inference patterns encoded by the DNN. First, it is proven that under some common conditions[2], a well-trained DNN usually just encodes a limited number of interactions *w.r.t.* a few sets of input variables. More crucially, let us randomly mask an input sample x in different ways to generate an exponential number of masked samples. *It is proven that people can use just a few interactions to accurately approximate the DNN's outputs on all these masked samples.* Thus, these few interactions are referred to as **interaction primitives**.

Despite the aforementioned theorems, this study does not yet deem the above interactions as faithful primitives of DNNs. The core problem is that existing interaction extraction methods cannot theoretically guarantee the generalization (transferability) of the interactions, *e.g.*, ensuring to extract

---

[*]Equal contribution.

[†]Quanshi Zhang is the corresponding author. He is with the Department of Computer Science and Engineering, the John Hopcroft Center, at the Shanghai Jiao Tong University, China.

[1]https://github.com/sjtu-xai-lab/generalizable-interaction

[2]Please see Appendix B for details.

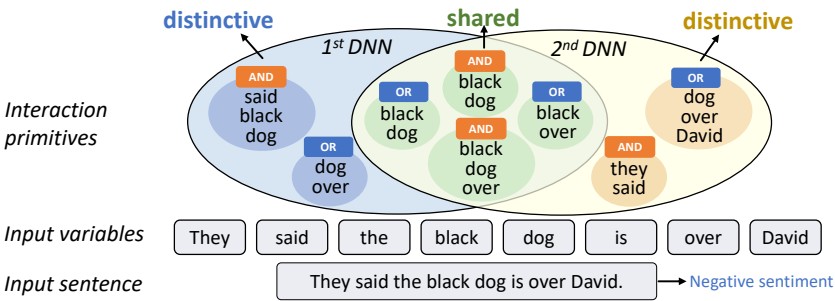

Figure 1: Distinctive and shared interactions. When we extract AND-OR interactions from two DNNs, AND interactions $S_1 = \{black, dog\}$ and $S_2 = \{black, dog, over\}$, and OR interactions $S_3 = \{black, dog\}$ and $S_4 = \{black, over\}$ are shared by two DNNs, while interactions, such as the AND interaction $S_5 = \{they, said\}$, are distinctive interactions encoded by a single DNN.

common interactions shared by different AI models. Interactions, which are not shared by different DNNs, may be perceived as out-of-distribution signals without clear meanings.

Therefore, in this study, we revisit the generalization of interactions. Specifically, we identify a clear mechanism that makes the existing method extract different interactions from the same DNN under different initialization states, which hurts the generalization power of interactions.

Thus, to address the generalization issue, we propose a new method for extracting generalizable interactions. A generalizable interaction is defined as Figure 1 shows. Given multiple DNNs trained for the same task and an input sample, if an interaction can be extracted from all these DNNs, then we consider it generalizable. Our method is designed to extract interactions with maximum generalization power. This approach ensures that if an interaction exhibits a significant impact on the output score for one DNN, it usually demonstrates noteworthy influence for other DNNs. We conducted experiments on various dataset. Experiments showed that our proposed method significantly improved the generalization power of the extracted interactions across different DNNs.

## 2 GENERALIZABLE INTERACTION PRIMITIVES ACROSS DNNS

### 2.1 PRELIMINARIES: EXPLAINING THE NETWORK OUTPUT WITH INTERACTION PRIMITIVES

Although there is no theory to guarantee that how to define concepts that fit well with human cognition, Li & Zhang (2023a) and Ren et al. (2024) still provided mathematical supports to explain why we can still use interactions between input variables as the primitives or concepts encoded by the DNN. Specifically, there are two types of interactions, *i.e.*, AND interactions and OR interactions.

**AND interactions.** Given a function $v : \mathbb{R}^n \to \mathbb{R}$, let us consider an input sample $\mathbf{x} = [x_1, x_2, \cdots, x_n]^\mathsf{T}$ with $n$ input variables indexed by $N = \{1, 2, ..., n\}$. Here, $v(\mathbf{x}) \in \mathbb{R}$ denotes the function output on $\mathbf{x}$[3]. Then, Ren et al. (2023b) have used the Harsanyi dividend (Harsanyi, 1963) $I_{\text{and}}(S|\mathbf{x})$ to quantify the numerical effect of the AND relationship between input variables in $S \subseteq N$, which is encoded by the function $v$. We consider this interaction as an *AND interaction*.

$$I_{\text{and}}(S|\mathbf{x}) := \sum_{T \subseteq S} (-1)^{|S|-|T|} v(\mathbf{x}_T). \tag{1}$$

where $I_{\text{and}}(\emptyset|\mathbf{x}) = v(\mathbf{x}_\emptyset)$, and $\mathbf{x}_T$ denotes a sample whose input variables in $N \setminus T$ are masked[4].

Each AND interaction $I_{\text{and}}(S|\mathbf{x})$ reveals the AND relationship between all variables in $S$. For instance, let us consider the slang term $S = \{x_3, x_4, x_5, x_6\}$ in the sentence "$x_1 = It, x_2 = is, x_3 = raining, x_4 = cats, x_5 = and, x_6 = dogs!$" as a toy example. The co-occurrence of four words forms the semantic concept of "heavy rain" and contributes a numerical effect $I_{\text{and}}(S|\mathbf{x})$ to the function output. Otherwise, the masking of any word $x_i \in S$ invalidates the semantic concept and eliminates the interaction effect, *i.e.*, obtaining $I_{\text{and}}(S|\mathbf{x}^{\text{masked}}) = 0$ on the masked sample.

---

[3] If the target function/model/network has a vectorized output, *e.g.*, a DNN for multi-category classification, we may set $v(\mathbf{x}) = \log \frac{p(y=y^{\text{truth}}|\mathbf{x})}{1-p(y=y^{\text{truth}}|\mathbf{x})}$ by following (Deng et al., 2022).

[4] We followed (Li & Zhang, 2023a) to obtain two discrete states for each input variable, *i.e.*, the masked and unmasked states. We simply masked each input variable $i \in N \setminus S$ using baseline values.

**OR interactions.** Analogously, we can also use the OR interaction to explain the function $v : \mathbb{R}^n \to \mathbb{R}$. To this end, (Zhou et al., 2023; Li & Zhang, 2023a) have defined the following OR interaction effect $I_{\text{or}}(S|\mathbf{x})$ to measure the OR interaction encoded by $v$. In particular, $I_{\text{or}}(\emptyset|\mathbf{x}) = v(\mathbf{x}_\emptyset)$.

$$I_{\text{or}}(S|\mathbf{x}) := -\sum_{T \subseteq S} (-1)^{|S|-|T|} v(\mathbf{x}_{N \setminus T}). \quad (2)$$

Each OR interaction $I_{\text{or}}(S|\mathbf{x})$ describes the OR relationship between all variables in $S$. Let us consider an input sentence "$x_1 = $ *This*, $x_2 = $ *movie*, $x_3 = $ *is*, $x_4 = $ *boring*, $x_5 = $ *and*, $x_6 = $ *disappointing*" for sentiment classification. Let us set $S = \{x_4, x_6\}$. The presence of any word in $S$ will contribute a negative sentiment effect $I_{\text{or}}(S|\mathbf{x})$ to the function output.

**Sparsity of interactions.** Theoretically, according to Equation (1), a function can encode at most $2^n$ different AND interactions *w.r.t.* all $2^n$ subsets $\forall S, S \subseteq N$. However, Ren et al. (2024) have proved that under some common conditions[2], most well-trained DNNs only encode a small set of AND interactions, denoted by $\Omega$, *i.e.*, only a few interactions $S \in \Omega$ have considerable effects $I_{\text{and}}(S|\mathbf{x})$. All other interactions have almost zero effects, *i.e.*, $I_{\text{and}}(S|\mathbf{x}) \approx 0$, which can be regarded as a set of negligible noise patterns.

It is worth noting that an OR interaction can be regarded as a specific AND interaction, if we inverse the definition of the masked state and the unmasked state of an input variable[5]. Thus, the proven sparsity of AND interactions can also indicate the conclusion that well-trained DNNs tend to encode a small number of OR interactions.

**Definition of interaction primitives.** Considering the above proven sparsity of interactions, we define an interaction primitive as a salient interaction. Formally, given a threshold $\tau$, the set of interaction primitives are defined as $\Omega = \{S \subseteq N : |I(S|\mathbf{x})| > \tau\}$.

**Theorem 1** (Universal matching theorem, proved by (Ren et al., 2024)). *As the corollary of the proven sparsity in Ren et al. (2024), the function's output on all $2^n$ masked samples $\{\mathbf{x}_S | S \subseteq N\}$ could be universally explained by the interaction primitives in $\Omega$, s.t., $|\Omega| \ll 2^n$, i.e., $\forall S \subseteq N, v(\mathbf{x}_S) = \sum_{T \subseteq S} I_{\text{and}}(T|\mathbf{x}) \approx \sum_{T \subseteq S: T \in \Omega} I_{\text{and}}(T|\mathbf{x})$.*

In particular, Theorem 1 shows that if we arbitrarily mask the input sample $\mathbf{x}$, we can get $2^n$ different masked samples[4], $\forall S, S \subseteq N$. Then, we can universally match the output of the function $v(\mathbf{x}_S)$ on all $2^n$ masked samples using only a few interaction primitives in $\Omega$.

## 2.2 Faithfulness problem with interaction-based explanations

**Basic setting of using AND-OR interactions to explain a DNN.** In this section, we consider to employ both AND interactions and OR interactions to explain the DNN's output. This is because the complexity of the representations in DNNs makes it difficult to rely solely on either AND interactions or OR interactions to faithfully explain true inference primitives encoded by the DNN.

To this end, we need to decompose the output of the DNN into two terms $v(\mathbf{x}) = v_{\text{and}}(\mathbf{x}) + v_{\text{or}}(\mathbf{x})$, so that we can use AND interactions to explain the term $v_{\text{and}}(\mathbf{x})$ and use OR interactions to explain the term $v_{\text{or}}(\mathbf{x})$. In this way, the first challenge is how to learn an appropriate decomposition of $v_{\text{and}}(\mathbf{x})$ and $v_{\text{or}}(\mathbf{x})$ that reveals intrinsic primitive interactions encoded by the DNN. We will discuss this challenge later.

No matter how we randomly decompose $v(\mathbf{x}) = v_{\text{and}}(\mathbf{x}) + v_{\text{or}}(\mathbf{x})$, Theorem 2 states that we can still use interactions to fit the DNN's outputs on $2^n$ randomly masked samples $\{\mathbf{x}_T | T \subseteq N\}$. Furthermore, according to the sparsity of interaction primitives in Section 2.1, we can obtain Proposition 1, *i.e.*, the $2^n$ network outputs on all masked samples can usually be approximated by a small number of AND interaction primitives in $\Omega^{\text{and}}$ and OR interaction primitives in $\Omega^{\text{or}}$, *s.t.*, $|\Omega^{\text{and}}|, |\Omega^{\text{or}}| \ll 2^n$.

**Theorem 2** (Universal matching theorem, proof in Appendix C). *Let us be given a DNN $v$ and an input sample $\mathbf{x}$. For each randomly masked sample $\mathbf{x}_T$, $T \subseteq N$, we obtain*

$$v(\mathbf{x}_T) = v_{\text{and}}(\mathbf{x}_T) + v_{\text{or}}(\mathbf{x}_T) = \sum_{S \subseteq T} I_{\text{and}}(S|\mathbf{x}_T) + \sum_{S \in \{S: S \cap T \neq \emptyset\} \cup \{\emptyset\}} I_{\text{or}}(S|\mathbf{x}_T). \quad (3)$$

---

[5]To compute $I_{\text{and}}(S|\mathbf{x})$, we use a baseline value $b_i$ and set $x_i = b_i$ to represent its masked state. If we consider $b_i$ variable as the presence of the variable, and consider the original value $x_i$ as its masked state (i.e., using $v(b_T)$ to represent $v(x_{N \setminus T})$ in Equation (2)), then $I_{\text{or}}(S|\mathbf{x})$ in Equation (2) can be formulated the same as the AND interaction in Equation (1).

**Proposition 1.** *The output of a well-trained DNN on all $2^n$ masked samples $\{\mathbf{x}_T | T \subseteq N\}$ could be universally approximated by the interaction primitives in $\Omega^{\mathrm{and}}$ and $\Omega^{\mathrm{or}}$, s.t., $|\Omega^{\mathrm{and}}|$, $|\Omega^{\mathrm{or}}| \ll 2^n$, i.e., $\forall T \subseteq N, v(\mathbf{x}_T) = \sum_{S \subseteq T} I_{\mathrm{and}}(S|\mathbf{x}_T) + \sum_{S \in \{S:S \cap T \neq \emptyset\} \cup \{\emptyset\}} I_{\mathrm{or}}(S|\mathbf{x}_T) \approx v(\mathbf{x}_\emptyset) + \sum_{\emptyset \neq S \subseteq T:S \in \Omega^{\mathrm{and}}} I_{\mathrm{and}}(S|\mathbf{x}_T) + \sum_{S \cap T \neq \emptyset:S \in \Omega^{\mathrm{or}}} I_{\mathrm{or}}(S|\mathbf{x}_T)$, where $v(\mathbf{x}_\emptyset) = v_{\mathrm{and}}(\mathbf{x}_\emptyset) + v_{\mathrm{or}}(\mathbf{x}_\emptyset)$.*

**Problems with the faithfulness of interactions.** Although the universal matching capacity proven in Theorem 2 is a quite significant advantage of AND-OR interactions, it is still not the ultimate guarantee for the faithfulness of the extracted interactions. To be precise, there is still no standard way to faithfully decompose the $v_{\mathrm{and}}(\mathbf{x})$ term and the $v_{\mathrm{or}}(\mathbf{x})$ term that reveal intrinsic primitive interactions encoded by the DNN, considering the following two challenges.

● *Challenge 1, ambiguous decomposition of $v_{\mathrm{and}}(\mathbf{x})$ and $v_{\mathrm{or}}(\mathbf{x})$ usually brings in considerable uncertainty in the extraction of interactions.* Let us take the following toy Boolean function as an example to illustrate the diversity of interactions, $f(\mathbf{x}) = x_1 \wedge x_2 \wedge x_3 + x_2 \wedge x_3 + x_3 \wedge x_4 + x_4 \vee x_5$, where $\mathbf{x} = [x_1, x_2, x_3, x_4, x_5]^{\mathsf{T}}$ and $x_i \in \{0, 1\}$. We have two ways to decompose $f(\mathbf{x})$. First, we can simply decompose $v_{\mathrm{and}}(\mathbf{x}) = x_1 \wedge x_2 \wedge x_3 + x_2 \wedge x_3 + x_3 \wedge x_4$ and $v_{\mathrm{or}}(\mathbf{x}) = x_4 \vee x_5$, then to explain $f(\mathbf{x})$ with an OR interaction $I_{\mathrm{or}}(S = \{4, 5\})$ and three AND interactions $I_{\mathrm{and}}(S = \{1, 2, 3\})$, $I_{\mathrm{and}}(S = \{2, 3\})$ and $I_{\mathrm{and}}(S = \{3, 4\})$. Alternatively, we can also use exclusively AND interactions to explain $f(\mathbf{x})$. Specifically, we can rewrite $v_{\mathrm{and}}(\mathbf{x}) = x_1 \wedge x_2 \wedge x_3 + x_2 \wedge x_3 + x_3 \wedge x_4 + x_4 \vee x_5 = x_1 \wedge x_2 \wedge x_3 + x_2 \wedge x_3 + x_3 \wedge x_4 + (x_4 + x_5 - x_4 \wedge x_5)$ and $v_{\mathrm{or}}(\mathbf{x}) = 0$, *w.r.t* $x_i \in \{0, 1\}$. Thus, the $v_{\mathrm{and}}$ term can be explained by a total of six AND interaction primitives. This is a typical case for diverse strategy of extracting interactions that are generated by different decompositions.

The aforementioned $f(\mathbf{x})$ is just an exceedingly simple function. In real-world applications, DNNs usually encode intricate AND-OR relationships among input variables, making it exceptionally challenging to formulate an explicit expression for the DNN function or to establish a definitive ground-truth decomposition of $v_{\mathrm{and}}(\mathbf{x})$ and $v_{\mathrm{or}}(\mathbf{x})$. Consequently, the diversity issue with interactions are ubiquitous and unavoidable.

● *Challenge 2, how to ensure the interaction primitives are generalizable.* It is commonly considered that generalizable primitives are usually transferable over different models trained for the same task, instead of being over-fitted by a single model. Thus, if an interaction primitive can be consistently extracted from different DNNs, then it can be considered as a faithful concept. Otherwise, non-generalizable (non-transferable) interactions do not appear as faithful concepts, even though they still satisfy the criteria of sparsity and universal matching in Theorem 2.

**Definition 1** (Transferability of interaction primitives). *Given $m$ different DNNs trained for the same task, $v^{(1)}, v^{(2)}, \ldots, v^{(m)}$, we use AND-OR interactions to explain the output score $v^{(i)}(\mathbf{x})$ of each DNN $v^{(i)}$ on the input sample $\mathbf{x}$. Let $\Omega^{\mathrm{and},(i)} = \{S \subseteq N : |I_{\mathrm{and}}^{(i)}(S|\mathbf{x})| > \tau^{(i)}\}$ and $\Omega^{\mathrm{or},(i)} = \{S \subseteq N : |I_{\mathrm{or}}^{(i)}(S|\mathbf{x})| > \tau^{(i)}\}$ denote a set of sparse AND interaction primitives and a set of sparse OR interaction primitives, respectively. Then, the set of generalizable AND and the set of generalizable OR interaction primitives for the $i$-th DNN, are defined as $\Omega_{\mathrm{shared}}^{\mathrm{and}} = \bigcap_{i=1}^{m} \Omega^{\mathrm{and},(i)}$ and $\Omega_{\mathrm{shared}}^{\mathrm{or}} = \bigcap_{i=1}^{m} \Omega^{\mathrm{or},(i)}$, respectively. The generalization power of AND and OR interactions of the $i$-th DNN, can be measured by $s_{\mathrm{and}}^{(i)} = |\Omega_{\mathrm{shared}}^{\mathrm{and}}|/|\Omega^{\mathrm{and},(i)}|$ and $s_{\mathrm{or}}^{(i)} = |\Omega_{\mathrm{shared}}^{\mathrm{or}}|/|\Omega^{\mathrm{or},(i)}|$, respectively.*

Definition 1 introduces the generalization power of interaction primitives. A larger value signifies higher transferability and, consequently, more generalizable interactive primitives.

## 2.3 EXTRACTING GENERALIZABLE INTERACTION PRIMITIVES

Neither of the aforementioned two challenges has been adequately tackled in previous interaction studies. In essense, interactions are determined by the decomposition of the network output $v(\mathbf{x}) = v_{\mathrm{and}}(\mathbf{x}) + v_{\mathrm{or}}(\mathbf{x})$. Thus, if we rewrite the decomposition as $v_{\mathrm{and}}(\mathbf{x}_T) = 0.5v(\mathbf{x}_T) + \gamma_T$ and $v_{\mathrm{or}}(\mathbf{x}_T) = 0.5v(\mathbf{x}_T) - \gamma_T$, then the learning of the best decomposition is equivalent to learning a set of $\{\gamma_T\}$. Here, the parameter $\gamma_T \in \mathbb{R}$ for a subset $T \subseteq N$ determines a specific decomposition between $v_{\mathrm{and}}(\mathbf{x}_T)$ and $v_{\mathrm{or}}(\mathbf{x}_T)$. Therefore, our goal is to learn the appropriate parameters $\{\gamma_T\}$ that reduce the aforementioned uncertainty of interaction primitives and boost their generalization power.

To this end, to alleviate the uncertainty of the interactions, the most intuitive approach is to learn the sparsest interactions, considering the principle of Occam's Razor, as follows. It is because the sparsest (or simplest) explanation is usually considered as the most faithful explanation.

$$\min_{\{\gamma_T\}} \|\mathbf{I}_{\text{and}}\|_1 + \|\mathbf{I}_{\text{or}}\|_1, \tag{4}$$

where $\mathbf{I}_{\text{and}} = [I_{\text{and}}(T_1|\mathbf{x}), \ldots, I_{\text{and}}(T_{2^n}|\mathbf{x})]^\mathsf{T}$, $\mathbf{I}_{\text{or}} = [I_{\text{or}}(T_1|\mathbf{x}), \ldots, I_{\text{or}}(T_{2^n}|\mathbf{x})]^\mathsf{T} \in \mathbb{R}^{2^n}$, $T_k \subseteq N$.

The above $\ell_1$ norm loss promotes the sparsity of both AND interactions and OR interactions.

### 2.3.1 ONLY ACHIEVING SPARSITY IS NOT ENOUGH

Although the sparsity can be used to reduce the uncertainty of interactions, the sparsity of interactions *w.r.t.* each single input sample obtained in Equation (4) does not fully solve the above two challenges. *First, Ren et al. (2023c) have found that the extraction of high-order interactions is usually sensitive to small noises in input variables*, where the order is defined as the number of input variables in $S$, *i.e.*, $\text{order}(S) = |S|$. It means that when different noises are added to the input samples, the algorithm may extract fully different high-order interactions[6]. Similarly, this will also hurt the generalization power of interaction primitives over different samples, when these samples contain similar sets of input variables.

*Second, optimizing the loss in Equation* (4) *may lead to diverse solutions.* Given different initial states, the loss in Equation (4) may learn two different sets of parameters $\{\gamma_T\}$ as two local minima with similar loss values, while the two sets of parameters $\{\gamma_T\}$ generate two different sets of interactions. We conducted experiments to illustrate this point. Given a pre-trained BERT model (Devlin et al., 2019) and an input sentence $\mathbf{x}$ on the SST-2 dataset for sentiment classification, we learned the parameters $\{\gamma_T\}$ to extract sparse interaction primitives[4]. In this experiment, we repeatedly extracted two sets of AND-OR interactions by applying two different sets of initialized parameters $\{\gamma_T\}$, which are denoted by $(\mathcal{A}^{\text{and}}, \mathcal{A}^{\text{or}})$ and $(\mathcal{B}^{\text{and}}, \mathcal{B}^{\text{or}})$. $\mathcal{A}^{\text{and}} = \{S \subseteq N : |I_{\text{and}}(S|\mathbf{x})| > \tau_{\mathcal{A}^{\text{and}}}\}$ denotes the set of AND interaction primitives extracted by a certain initialization of $\{\gamma_T\}$, where the parameter $\tau_{\mathcal{A}^{\text{and}}}$ was determined to ensure that each set selected the most salient $K = 100$ interactions. We used the transferability of interaction primitives in Definition 1, $\mathcal{S}_{\text{and}} = |\mathcal{A}^{\text{and}} \bigcap \mathcal{B}^{\text{and}}|/|\mathcal{A}^{\text{and}}|$ and $\mathcal{S}_{\text{or}} = |\mathcal{A}^{\text{or}} \bigcap \mathcal{B}^{\text{or}}|/|\mathcal{A}^{\text{or}}|$, to measure the diversity of interactions caused by the different parameter initializations. Table 2 shows that given different initial states, optimizing the loss in Equation (4) usually extracted two dramatically different sets of AND-OR interactions with only 21% overlap. Figure 12 further shows the top 5 AND-OR interaction primitives extracted from BERT model on the same input sentence, which illustrates that given different initial states, the loss in Equation (4) would learn different AND-OR interactions.

*Third, prioritizing sparsity cannot guarantee high generalization power across different models.* Since a DNN may simultaneously learn common knowledge shared by different DNNs and be overfitted to some out-of-the-distribution patterns, different DNNs may only share partial interaction primitives. We believe that the shared common interactions are more faithful, so the transferability is another way to guarantee the generalization power of interaction primitives. Therefore, we hope to formulate and extract common interactions that are generalizable through different DNNs.

We conducted experiments to illustrate the difference between interactions extracted from two DNNs by using Equation (4). We used BERT$_{\text{BASE}}$ $v_{\text{base}}$ and BERT$_{\text{LARGE}}$ $v_{\text{large}}$ (Devlin et al., 2019) for the task of sentiment classification. Specifically, given an input sentence $\mathbf{x}$, we learned two sets of parameters $\{\gamma_T^{\text{base}}\}$ and $\{\gamma_T^{\text{large}}\}$ for the BERT-base model and the BERT-large model, respectively. Then we extracted two sets of AND-OR interactive concepts $(\Omega^{\text{and,base}}, \Omega^{\text{or,base}})$ and $(\Omega^{\text{and,large}}, \Omega^{\text{or,large}})$, respectively. Subsequently, we computed the transferability of the extracted interaction primitives according to Definition 1. Figure 3(a) shows that the transferability of the extracted AND-OR interaction primitives was much lower than interactions proposed in this study.

### 2.3.2 EXTRACTING GENERALIZABLE INTERACTIONS

As discussed above, the sparsity alone is not enough to tackle the aforementioned challenges. Therefore, in this study, we propose to use the generalization power as a straightforward purpose, to

---

[6]Please see Appendix D for details.

boost the faithfulness of interactions. Meanwhile, the sparsity of interactions is also supposed to be guaranteed. Given a total of $m$ DNNs $v^{(1)}, v^{(2)}, \ldots, v^{(m)}$ trained for the same task, the objective of extracting generalizable interactions shared by the $m$ DNNs is revised from Equation (4), as follows.

$$\min_{\{\gamma_T^{(1)}, \ldots, \gamma_T^{(m)}\}} \|\mathrm{rowmax}(\mathbb{I}_{\mathrm{and}})\|_1 + \|\mathrm{rowmax}(\mathbb{I}_{\mathrm{or}})\|_1, \tag{5}$$

where $\mathbb{I}_{\mathrm{and}} = \left[ \mathbf{I}_{\mathrm{and}}^{(1)} \ \mathbf{I}_{\mathrm{and}}^{(2)} \ \ldots \ \mathbf{I}_{\mathrm{and}}^{(m)} \right] \in \mathbb{R}^{2^n \times m}$ and $\mathbb{I}_{\mathrm{or}} = \left[ \mathbf{I}_{\mathrm{or}}^{(1)} \ \mathbf{I}_{\mathrm{or}}^{(2)} \ \ldots \ \mathbf{I}_{\mathrm{or}}^{(m)} \right] \in \mathbb{R}^{2^n \times m}$. $\mathbf{I}_{\mathrm{and}}^{(i)} = [I_{\mathrm{and}}^{(i)}(T_1|\mathbf{x}), \ldots, I_{\mathrm{and}}^{(i)}(T_{2^n}|\mathbf{x})]^\mathsf{T} \in \mathbb{R}^{2^n}$ and $\mathbf{I}_{\mathrm{or}}^{(i)}$ represent the all $2^n$ AND-OR interactions extracted from the $i$-th DNN, $T_k \subseteq N$. The matrix operator $\mathrm{rowmax}()$ computes the $\ell_\infty$ norm of each row within the matrix, $i.e.$, $\mathrm{rowmax}(\mathbb{I}_{\mathrm{and}}) = [\|\mathbb{I}_{\mathrm{and}}[1,:]\|_\infty, \ldots, \|\mathbb{I}_{\mathrm{and}}[2^n,:]\|_\infty]^\mathsf{T} \in \mathbb{R}^{2^n}$. For each specific subset of variables $T_k \subseteq N$, the $\mathrm{rowmax}()$ operation returns the most salient interaction strength over all $m$ interactions from the $m$ DNNs. Please see Appendix F for more discussion on the matrix $\mathbb{I}_{\mathrm{and}}$.

Unlike Equation (4), the revised loss in Equation (5) only penalizes the most salient interactions over all $m$ interactions extracted from $m$ DNNs, with respect to each subset $T_k \subseteq N$. This loss function ensures that if a DNN encodes a strong interaction $w.r.t.$ the set $T_k$, then we can also extract the same interaction $w.r.t.$ $T_k$ from the other $m-1$ DNNs without a penalty. The $\ell_1$ norm also makes that the $m$ DNNs share similar sets of sparse interactions. Considering the sparsity of interactions, for most subset $T_k$, the effect $I_{\mathrm{and/or}}^{(i)}(T_k|\mathbf{x})$ is supposed to keep almost zero on all $m$ DNNs.

Just like in Equation (4), we decompose the output of the $i$-th DNN as $v_{\mathrm{and}}^{(i)}(\mathbf{x}_T) = 0.5v^{(i)}(\mathbf{x}_T) + \gamma_T^{(i)}$ and $v_{\mathrm{or}}^{(i)}(\mathbf{x}_T) = 0.5v^{(i)}(\mathbf{x}_T) - \gamma_T^{(i)}$ to compute two vectors of AND-OR interactions, $\mathbf{I}_{\mathrm{and}}^{(i)}$ and $\mathbf{I}_{\mathrm{or}}^{(i)}$.

**Redundancy of interactions.** However, it is important to emphasize that *only penalizing the largest interaction among the $m$ DNNs* in Equation (5) still faces the redundancy problem. Specifically, for each $i$-th DNN, we compute a total of $2^n$ AND interactions and $2^n$ OR interactions $w.r.t.$ different subsets $T \subseteq N$. Some of these $2^{n+1}$ interactions, denoted by the set $\Omega_{\mathrm{max}}^{(i)}$, are selected by the loss in Equation (5) as the most salient interactions over $m$ DNNs, while the set of other unselected interactions are denoted by $\Omega_{\mathrm{others}}^{(i)} = \{T \subseteq N\} \setminus \Omega_{\mathrm{max}}^{(i)}$. Then, the redundancy problem is caused by a short-cut solution to the loss minimization in Equation (5), $i.e.$, using unselected not-so-salient interactions in $\Omega_{\mathrm{others}}^{(i)}$ to represent numerical effects of selected interactions $\Omega_{\mathrm{max}}^{(i)}$, as discussed in Challenge 1 in Section 2.2. As a short-cut solution to Equation (5), this may also reduces the strength of the penalized salient interactions in Equation (5), but generates lots of redundant interacitons.

Therefore, we revise the loss in Equation (5) to add penalties on unselected interactions to avoid the short-cut solution with a coefficient of $\alpha$, as follows[7].

$$\min_{\{\gamma_T^{(1)}, \ldots, \gamma_T^{(m)}\}} \left( \|\mathrm{rowmax}(\mathbb{I}_{\mathrm{and}})\|_1 + \|\mathrm{rowmax}(\mathbb{I}_{\mathrm{or}})\|_1 \right) + \alpha \left( \|\mathbb{I}_{\mathrm{and}}\|_1 + \|\mathbb{I}_{\mathrm{or}}\|_1 \right), \tag{6}$$

where $\alpha \in [0, 1]$ is a positive scalar. We extend the notation of the $\ell_1$ norm $\| \cdot \|_1$ to represent the sum of the absolute values of all elements in a given vector or matrix. It is worth noting that the generalization power of interactions is guaranteed by the $\mathrm{rowmax}()$ function in Equation (5), which assigns much higher penalties to non-generalizable interactions than generalizable interactions.

**Sharing decomposition between DNNs.** Optimizing Equation (6) is challenging[8]. To address this challenge, we introduce a set of strategies to facilitate the optimization process. We assume that when all $m$ DNNs are sufficiently trained, these DNNs tend to have similar decompositions of AND interactions and OR interactions, $i.e.$, obtaining similar parameters, $\forall T \subseteq N, \gamma_T^{(1)} \approx \gamma_T^{(2)} \approx \cdots \approx \gamma_T^{(m)}$. To achieve this, we introduce two types of parameters for $\gamma_T^{(i)}$, $\gamma_T^{(i)} = \bar{\gamma}_T + \hat{\gamma}_T^{(i)}$, where $\bar{\gamma}_T$ represents the common decomposition shared by all DNNs, and $\hat{\gamma}_T^{(i)}$ represents the decomposition specific to each $i$-th DNN. We constrain the significance of the unshared decomposition by using a bound $|\hat{\gamma}_T^{(i)}| < \tau_\gamma^{(i)}$, where $\tau_\gamma^{(i)} = 0.5 \cdot \mathbb{E}_{\mathbf{x}}[|v^{(i)}(\mathbf{x}) - v^{(i)}(\mathbf{x}_\emptyset)|]$. During the training process, if $|\hat{\gamma}_T^{(i)}| > \tau_\gamma^{(i)}$, then we set $\hat{\gamma}_T^{(i)} = \tau_\gamma^{(i)} \cdot \mathrm{sign}(\hat{\gamma}_T^{(i)})$.

---

[7]Please see Appendix F for a detailed explanation of Equation (6).

[8]Note that regardless of whether Equation (6) is optimized to the optimal solution, theoretically, the extracted AND-OR interactions can still satisfy the property of universal matching in Theorem 2.

**Modeling noises.** Furthermore, we have identified a potential limitation in the definition of the interactions, *i.e.*, the sensitivity to noise. Let us assume that the output of the $i$-th DNN has a small noise. We represent such noises by adding a small Gaussian noise $\epsilon_T \sim \mathcal{N}(0, \sigma^2)$ to the network output $v'^{(i)}_{\text{and}}(\mathbf{x}_T) = v^{(i)}_{\text{and}}(\mathbf{x}_T) + \epsilon^{(i)}_T$. In this case, we can derive that $I'^{(i)}_{\text{and}}(T) = I^{(i)}_{\text{and}}(T) + \sum_{T' \subseteq T} (-1)^{|T|-|T'|} \epsilon^{(i)}_{T'}$. We prove that the variance of $I'^{(i)}_{\text{and}}(T)$ caused by the Gaussian noises is $\mathbb{E}_{\epsilon_T \sim \mathcal{N}(0, \sigma^2)}[I'^{(i)}_{\text{and}}(T) - E_{\forall S, \epsilon_S \sim \mathcal{N}(0, \sigma^2)} I'^{(i)}_{\text{and}}(S)]^2 = 2^{|T|} \sigma^2$ (please see Appendix D for details). Similarly, the variance of $I'^{(i)}_{\text{or}}(T)$ is also $2^{|T|} \sigma^2$ for OR interactions. It means that the variance/instability of interactions increases exponentially with the order of the interaction $|T|$.

Therefore, we propose to directly learn the error term $\epsilon^{(i)}_T$ based on Equation (6) to remove tiny noisy signals, which are unavoidable in real data but cannot be modeled as AND-OR interactions, *i.e.*, setting $v^{(i)}(\mathbf{x}_T) = v^{(i)}_{\text{and}}(\mathbf{x}_T) + v^{(i)}_{\text{or}}(\mathbf{x}_T) + \epsilon^{(i)}_T$, in order to enhance the robustness of our interaction extraction process. The error term is constrained to a small range $|\epsilon^{(i)}_T| < \tau^{(i)}_\epsilon$, subject to $\tau^{(i)}_\epsilon = 0.02 \cdot |v^{(i)}(\mathbf{x}) - v^{(i)}(\mathbf{x}_\emptyset)|$. During the training process, if $|\epsilon^{(i)}| > \tau^{(i)}_\epsilon$, then we set $|\epsilon^{(i)}| = \tau^{(i)}_\epsilon \cdot \text{sign}(\epsilon^{(i)}_T)$.

Then, we conducted experiments to examine whether the extracted AND-OR interactions could still accurately explain the network output, when removing the error term. We followed experimental settings in Section 3 to extract interactions on both BERT$_{\text{BASE}}$ and BERT$_{\text{LARGE}}$ models. We computed the matching error $e(\mathbf{x}_T) = |v(\mathbf{x}_T) - v^{\text{approx}}(\mathbf{x}_T)|$, where $v^{\text{approx}}(\mathbf{x}_T)$ was the network output approximated by all interactions based on Theorem 2. Figure 13 shows matching errors of all masked samples *w.r.t.* all subsets $T \subseteq N$, when we sorted the network outputs for all $2^n$ masked samples in a descending order. It shows that the real network output was well approximated by interactions.

# 3 EXPERIMENT

In this section, we conducted experiments to verify the sparsity and generalization power of the interaction primitives extracted by our proposed method on the following three tasks.

*Task1: sentiment classification with language models.* We jointly extracted two sets of AND-OR interaction primitives from the BERT$_{\text{BASE}}$ model and the BERT$_{\text{LARGE}}$ model (Devlin et al., 2019) by following Equation (6). We finetuned the pre-trained BERT$_{\text{BASE}}$ model and the BERT$_{\text{LARGE}}$ model on the SST-2 dataset (Socher et al., 2013) for sentiment classification. For each input sentence $\mathbf{x}$ containing $n$ tokens[9], we analyzed the log-odds output of the ground-truth label, *i.e.*, $v(\mathbf{x}) = \log \frac{p(y=y^{\text{truth}}|\mathbf{x})}{1-p(y=y^{\text{truth}}|\mathbf{x})}$ by following (Deng et al., 2022).

*Task2: dialogue task with large language models.* We extracted two sets of AND-OR interaction primitives from the pre-trained LLaMA model (Touvron et al., 2023) and OPT-1.3B model (Zhang et al., 2022b). We explained the DNNs' outputs on the SQuAD dataset (Rajpurkar et al., 2016). We took the first several words of each document in the dataset as the input of a DNN, and let the DNN predict the next word. For each input sentence $\mathbf{x}$ containing $n$ words[9], we analyzed the log-odds output of the $(n+1)$-th word that was associated with the highest probability by the DNN, $y^{\text{max}}$, *i.e.*, $v(\mathbf{x}) = \log \frac{p(y=y^{\text{max}}|\mathbf{x})}{1-p(y=y^{\text{max}}|\mathbf{x})}$.

*Task3: image classification task with vision models.* We extracted two sets of AND-OR interaction primitives from the ResNet-20 model (He et al., 2016) and the VGG-16 model (Simonyan & Zisserman, 2015), which were trained on the MNIST dataset (LeCun, 1998). These models were trained to classify the digit "3" from other digits. In practice, considering the $2^n$ computational complexity, we have followed settings in (Li & Zhang, 2023b), who labeled a few important input patches in the image as input variables. For each input image $\mathbf{x}$ containing $n$ patches[9], we analyzed the scalar output before the softmax layer corresponding to the digit "3."

**Sparsity of the extracted primitives.** We aggregated all AND-OR interactions from various samples, and draw their strength in a descending order in Figure 2. This figure compares the curve of interaction strength $|I(S)|$, $S \subseteq N$ between our extracted interactions, the traditional interactions (Li

---

[9]Please see Appendix O for details.

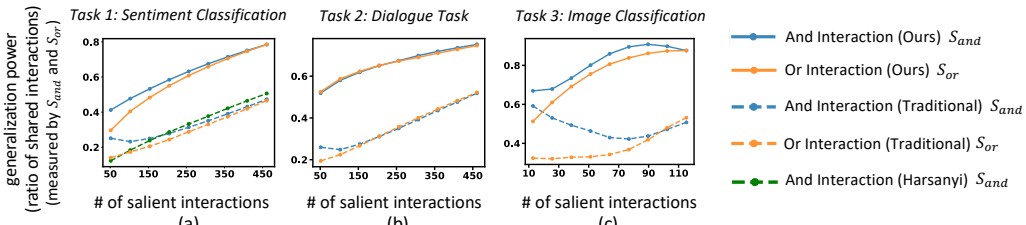

Figure 2: Strength of AND-OR interactions $\log|I(S|\mathbf{x})|$ over different samples in a descending order. All interactions above the dash line had much more significant effect (shown in a log space) and were considered as salient interactions.

Figure 3: Generalization power (measured by $s_{\text{and}}$ and $s_{\text{or}}$) of the extracted primitives interactions.

& Zhang, 2023b)[10](namely, *Traditional*) and the original Harsanyi interactions (Ren et al., 2023a) (namely, *Harsanyi*). The competing method (Li & Zhang, 2023b) (*Traditional* in Figure 2) extracts the sparest interactions according to Equation (4), and the original Harsanyi interactions (Ren et al., 2023a) (*Harsanyi* in Figure 2) extracts interactions according to Equation (1)[11]. We found that most of the interactions had negligible effect. Although the proposed method reduced the sparsity a bit, the extracted interactions were still sparse enough to be considered as primitive inference patterns. For each DNN, we further set a threshold $\tau^{(i)}$ to collect a set of salient interactions from this DNN as interaction primitives, *i.e.*, $\tau^{(i)} = 0.05 \cdot \max_S |I(S|\mathbf{x})|$.

**Generalization power of the extracted interaction primitives.** We took the most salient $k$ interactions from each $i$-th DNN as the set of AND-OR interaction primitives, *i.e.*, $|\Omega^{\text{and, (i)}}| = |\Omega^{\text{or, (i)}}| = k, i \in \{1, 2\}$. We used the metric $s_{\text{and}}$ and $s_{\text{or}}$ in Definition 1 to measure the generalization power of interactions extracted from two DNNs. Figure 3 shows the generalization power of interactions when we computed $s_{\text{and}}$ and $s_{\text{or}}$ based on different numbers $k$ of most salient interactions. We found that the set of AND-OR interactions extracted from the proposed method exhibited higher generalization power than interactions extracted from the traditional method.

**Low-order interaction primitives are more stable.** Furthermore, we compared the ratio of shared interactions of different orders, *i.e.*, order$(S) = |S|$. For interactions of each order $o$, we computed the overall strength of all positive interactions and that of all negative interactions of the $i$-th DNN, which were shared by other DNNs, as $Shared^{+,(i)}(o) = \sum_{\text{op}\in\{\text{and,or}\}} \sum_{S\in\Omega^{\text{op},(i)}_{\text{shared}},|S|=o} \max(0, I^{(i)}_{\text{op}}(S|\mathbf{x}))$

and $Shared^{-,(i)}(o) = \sum_{\text{op}\in\{\text{and,or}\}} \sum_{S\in\Omega^{\text{op},(i)}_{\text{shared}},|S|=o} \min(0, I^{(i)}_{\text{op}}(S|\mathbf{x}))$, respectively. Besides, $All^{+,(i)}(o) = \sum_{\text{op}\in\{\text{and,or}\}} \sum_{S\in\Omega^{\text{op},(i)},|S|=o} \max(0, I^{(i)}_{\text{op}}(S|\mathbf{x}))$ and $All^{-,(i)}(o) = \sum_{\text{op}\in\{\text{and,or}\}} \sum_{S\in\Omega^{\text{op},(i)},|S|=o} \min(0, I^{(i)}_{\text{op}}(S|\mathbf{x}))$ denote the overall strength of salient positive interactions and that of salient negative interactions, respectively. In this way, Figure 4 reports $(Shared^{+,(i)}, Shared^{-,(i)})$ and $(All^{+,(i)}, All^{-,(i)})$ for different orders within the three tasks. It shows that low-order interactions were more likely to be shared by different DNNs than high-order interactions. Besides, Figure 4 further shows a higher ratio of interactions extracted by the proposed method were shared by different DNNs than interactions extracted by the traditional method. In particular, the high similarity of interactions between ResNet-20 and VGG-16 shows that although two DNNs for the same tasks had fully different architectures, there probably existed a set of ultimate interactions for a task, and different well-optimized DNNs were likely to converge to such interactions.

**Visualization of the shared interaction primitives across different DNNs.** We also visualize the shared and distinctive interaction primitives in Figure 5. This figure shows that generalizable interac-

---

[10]In the implementation of the competing method, we also learned an additional error term $\epsilon^{(i)}_T$ to remove small noises, just like in Section 2.3.2, to enable fair comparisons.

[11]Please see Appendix H for more details.

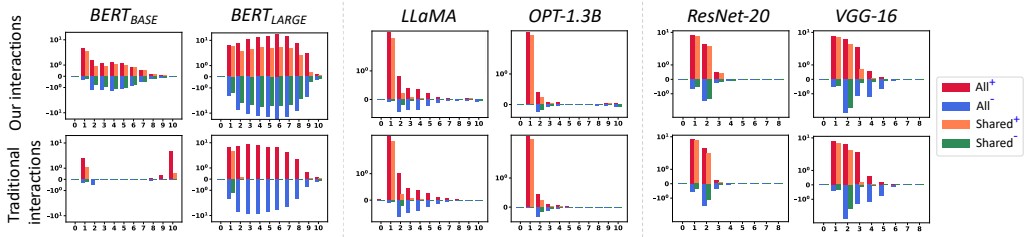

Figure 4: Overall interactions and shared interactions. The red and black bars show the overall strength of positive interactions $All^{+,(i)}(o)$ and that of negative interactions $All^{-,(i)}(o)$ of each $o$-th order. The orange and green bars indicate the strength of positive interactions that are shared by the other DNN $Shared^{+,(i)}(o)$ and that of the shared negative interactions $Shared^{-,(i)}(o)$, respectively.

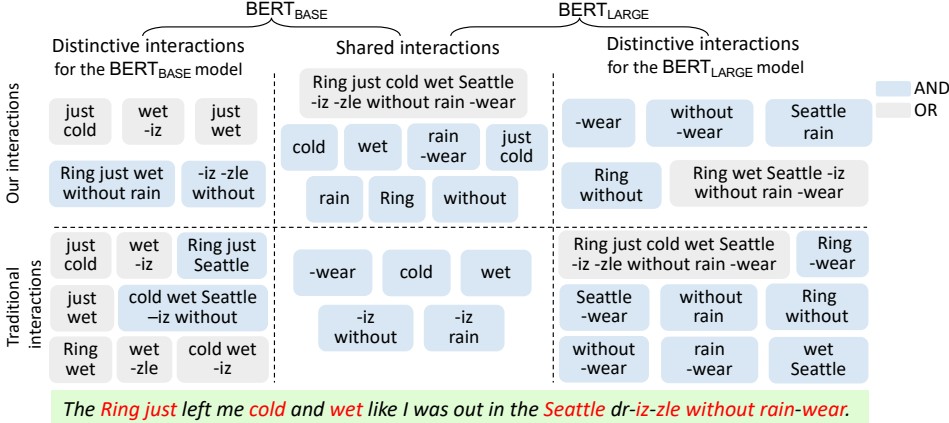

Figure 5: Visualization of the shared and distinctive interaction primitives across different DNNs. We selected some of salient interactions from the most salient $k = 50$ AND-OR interactions in each DNN. The black and gray color show the AND interactions and the OR interactions, respectively. The left and right column show the distinctive interactions extracted from the BERT$_{BASE}$ model and the BERT$_{LARGE}$ model, respectively. The middle column shows the shared interactions extracted from both models. Please see Appendix J for more interactions.

tions shared by different models can be regarded as more reliable concepts, which consistently contribute salient interaction effects to the output of different DNNs. In comparison, non-generalizable interactions, which are sometimes over-fitted by a single model, may appear as out-of-distribution features. From this pespective, we consider generalizable interactions as relatively faithful concepts that often have a significant impact on the inference of DNNs. Figure 5 further shows that, our method extracted much more shared interactions than the traditional interaction-extraction method, which shows that our method could obtain more stable explanation of the inference logic of a DNN. It is because the interactions shared by different DNNs were usually considered more faithful.

## 4 CONCLUSION

In this paper, we proposed a method to extract generalizable interaction primitives. The sparsity and universal-matching property of interactions provide lots of evidence to faithfully explain DNNs with interactions. Thus, in this paper, we propose to further improve the generalization power of interactions, which adds the last piece of the puzzle of interaction primitives. Compared to traditional interactions, interactions shared by different DNNs are more likely to be the underlying primitives that shape the DNN's output. Furthermore, the extraction of interaction primitives also contributes to real applications. For example, it can assist in learning optimal baseline values for Shapley values (Ren et al., 2023b) and explaining the representation limits of Bayesian networks (Ren et al., 2023c). In addition, the extraction of generalizable interaction primitives shared by different DNNs provide a new perspective to formulating the out-of-distribution (OOD) features. Previous studies usually treated an entire sample as an OOD sample, whereas our work redefines the OOD problem at the level of detailed interactions, *i.e.*, unshared interactions can be regarded as OOD information.

## ACKNOWLEDGMENTS

This work is partially supported by the National Science and Technology Major Project (2021ZD0111602), the National Nature Science Foundation of China (62276165,92370115), Shanghai Natural Science Foundation (21JC1403800,21ZR1434600).

## ETHICS STATEMENT

This paper aims to extract generalizable interaction primitives that are shared by different DNNs. This paper utilizes publicly released datasets which have been widely accepted by the machine learning community. This paper does not involve human subjects and does not include potentially harmful insights, methods, or applications. The paper also does not involve discrimination/bias/fairness issues, as well as privacy and security issues. There are no ethical issues with this paper.

## REPRODUCIBILITY STATEMENT

We provide proofs for the theoretical results of this study in Appendix C to D. We also provide experimental details in Section 3 and Appendix O.

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

## A    PREVIOUS LITERATURE OF USING INTERACTIONS TO EXPLAIN DNNS

Explaining the knowledge encoded by DNNs is one of the ultimate goals of explainable AI but presents significant challenges. For instance, some studies employed visualization techniques to show the patterns learned by a DNN (Simonyan et al., 2013; Yosinski et al., 2015), and some focused on extracting feature vectors that may be associated with semantic concepts (Simonyan et al., 2013), while some studies learned feature vectors potentially related to concepts (Kim et al., 2018). Dravid et al. (2023) identified convolutional kernels in different DNNs that expressed similar concepts.

However, theoretically, whether the knowledge or the complex inference logic of a DNN can be faithfully represented as symbolic primitive inference patterns still presents significant challenges. Up to now, there is still no universally accepted definition of the knowledge, as it encompasses various aspects of cognitive science, neuroscience, and mathematics. However, if we ignore cognitive and neuroscience aspects, Ren et al. (2023a) have proposed to quantify interactions between input variables encoded by the DNN, to explain the knowledge in the DNN. More crucially, Ren et al. (2024) have derived several theorems as mathematical evidence of considering such interactions as primitive inference patterns encoded by a DNN. Specifically, Ren et al. (2024) proved that DNNs usually only encoded a small number of interactions, under some common conditions[2]. Besides, these interactions can universally explain the DNN's output score on any arbitrary masked input samples. Li & Zhang (2023b) further discovered the discriminative power of certain interactions.

Besides, Deng et al. (2024) found that different attribution scores estimated by different attribution methods could all be represented as a combination of different interactions. Zhang et al. (2022a) used interactions to explain the mechanism of different methods of boosting adversarial transferability. Ren et al. (2023b) used interactions to define the optimal baseline value for computing Shapley values. Deng et al. (2022) found that for most DNNs it was difficult to learn interactions with median number of input variables, and it was discovered that DNNs and Bayesian neural networks were unlikely to model complex interactions with many input variables (Ren et al., 2023c; Liu et al., 2024). Zhou et al. (2024) used the generalization power of different interactions to explain the generalization power of DNNs.

## B    THE CONDITIONS FOR UNIVERSAL MATCHING OF THE DNN OUTPUT

Ren et al. (2024) have proved that a well-trained DNN usually just encodes a limited number of interactions. More importantly, one can just use a few interactions to approximate the DNN's outputs on all $2^n$ masked samples $\{\mathbf{x}_S | S \subseteq N\}$, under the following three common assumptions. (1) The high order derivatives of the DNN output with respect to the input variables are all zero. (2) The DNN works well on the masked samples, and yield higher confidence when the input sample is less masked. (3) The confidence of the DNN does not drop significantly on the masked samples. With these natural assumptions, a well-trained DNN's output on all $2^n$ masked samples can be universally approximated by the sum of just a few salient interactions in $\Omega$, $s.t., |\Omega| \ll 2^n$.

## C    PROOF OF THEOREMS

**Theorem 2** (Universal matching theorem) Let us be given a DNN $v$ and an input sample $\mathbf{x}$. For each randomly masked sample $\mathbf{x}_T, T \subseteq N$, we obtain

$$v(\mathbf{x}_T) = v_{\text{and}}(\mathbf{x}_T) + v_{\text{or}}(\mathbf{x}_T) = \sum_{S \subseteq T} I_{\text{and}}(S|\mathbf{x}_T) + \sum_{S \in \{S : S \cap T \neq \emptyset\} \cup \{\emptyset\}} I_{\text{or}}(S|\mathbf{x}_T). \quad (7)$$

*Proof.* **(1) Universal matching theorem of AND interactions.**

Ren et al. (2023a) have used the Haranyi dividend (Harsanyi, 1963) $I_{\text{and}}(S|\mathbf{x})$ to state the universal matching theorem of AND interactions. The output of a well-trained DNN on all $2^n$ masked samples $\{\mathbf{x}_T | T \subseteq N\}$ could be universally explained by the all interaction primitives in $T \subseteq N$, *i.e.*, $\forall T \subseteq N, v_{\text{and}}(\mathbf{x}_T) = \sum_{S \subseteq T} I_{\text{and}}(S|\mathbf{x})$.

Specifically, the AND interaction is defined as $I_{\text{and}}(S|\mathbf{x}) := \sum_{L \subseteq S} (-1)^{|S|-|L|} v_{\text{and}}(\mathbf{x}_L)$ in Equation (1). To compute the sum of AND interactions $\forall T \subseteq N, \sum_{S \subseteq T} I_{\text{and}}(S|\mathbf{x}) =$

$\sum_{S \subseteq T} \sum_{L \subseteq S} (-1)^{|S|-|L|} v_{\text{and}}(\mathbf{x}_L)$, we first exchange the order of summation of the set $L \subseteq S \subseteq T$ and the set $S \supseteq L$. That is, we compute all linear combinations of all sets $S$ containing $L$ with respect to the model outputs $v_{\text{and}}(\mathbf{x}_L)$ given a set of input variables $L$, *i.e.*, $\sum_{S:L \subseteq S \subseteq T} (-1)^{|S|-|L|} v_{\text{and}}(\mathbf{x}_L)$. Then, we compute all summations over the set $L \subseteq T$.

In this way, we can compute them separately for different cases of $L \subseteq S \subseteq T$. In the following, we consider the cases (1) $L = S = T$, and (2) $L \subseteq S \subseteq T, L \neq T$, respectively.

(1) When $L = S = T$, the linear combination of all subsets $S$ containing $L$ with respect to the model output $v_{\text{and}}(\mathbf{x}_L)$ is $(-1)^{|T|-|T|} v_{\text{and}}(\mathbf{x}_L) = v_{\text{and}}(\mathbf{x}_L)$.

(2) When $L \subseteq S \subseteq T, L \neq T$, the linear combination of all subsets $S$ containing $L$ with respect to the model output $v_{\text{and}}(\mathbf{x}_L)$ is $\sum_{S:L \subseteq S \subseteq T} (-1)^{|S|-|L|} v_{\text{and}}(\mathbf{x}_L)$. For all sets $S : T \supseteq S \supseteq L$, let us consider the linear combinations of all sets $S$ with number $|S|$ for the model output $v_{\text{and}}(\mathbf{x}_L)$, respectively. Let $m := |S|-|L|, (0 \leq m \leq |T|-|L|)$, then there are a total of $C_{|T|-|L|}^m$ combinations of all sets $S$ of order $|S|$. Thus, given $L$, accumulating the model outputs $v_{\text{and}}(\mathbf{x}_L)$ corresponding to all $S \supseteq L$, then $\sum_{S:L \subseteq S \subseteq T} (-1)^{|S|-|L|} v_{\text{and}}(\mathbf{x}_L) = v_{\text{and}}(\mathbf{x}_L) \cdot \underbrace{\sum_{m=0}^{|T|-|L|} C_{|T|-|L|}^m (-1)^m}_{=0} = 0$.

Please see the complete derivation of the following formula.

$$
\begin{aligned}
\sum_{S \subseteq T} I_{\text{and}}(S|\mathbf{x}_T) &= \sum_{S \subseteq T} \sum_{L \subseteq S} (-1)^{|S|-|L|} v_{\text{and}}(\mathbf{x}_L) \\
&= \sum_{L \subseteq T} \sum_{S:L \subseteq S \subseteq T} (-1)^{|S|-|L|} v_{\text{and}}(\mathbf{x}_L) \\
&= \underbrace{v_{\text{and}}(\mathbf{x}_T)}_{L=T} + \sum_{L \subseteq T, L \neq T} v_{\text{and}}(\mathbf{x}_L) \cdot \underbrace{\sum_{m=0}^{|T|-|L|} C_{|T|-|L|}^m (-1)^m}_{=0} \\
&= v_{\text{and}}(\mathbf{x}_T).
\end{aligned}
\tag{8}
$$

**(2) Universal matching theorem of OR interactions.**

According to the definition of OR interactions in Section 2.1, we will derive that $\forall T \subseteq N, v_{\text{or}}(\mathbf{x}_T) = \sum_{S \in \{S:S \cap T \neq \emptyset\} \cup \{\emptyset\}} I_{\text{or}}(S|\mathbf{x}_T) = I_{\text{or}}(\emptyset|\mathbf{x}_T) + \sum_{S:S \cap T \neq \emptyset} I_{\text{or}}(S|\mathbf{x}_T), s.t., I_{\text{or}}(\emptyset|\mathbf{x}_T) = v_{\text{or}}(\mathbf{x}_\emptyset)$.

Specifically, the OR interaction is defined as $I_{\text{or}}(S|\mathbf{x}) := -\sum_{L \subseteq S} (-1)^{|S|-|L|} v_{\text{or}}(\mathbf{x}_{N \setminus L})$ in Equation (2). Similar to the above derivation of the universal matching theorem of AND interactions, to compute the sum of OR interactions $\forall T \subseteq N, \sum_{S:S \cap T \neq \emptyset} I_{\text{or}}(S|\mathbf{x}_T) = \sum_{S:S \cap T \neq \emptyset} \left[ -\sum_{L \subseteq S} (-1)^{|S|-|L|} v_{\text{or}}(\mathbf{x}_{N \setminus L}) \right]$, we first exchange the order of summation of the set $L \subseteq S \subseteq N$ and the set $S : S \cap T \neq \emptyset$. That is, we compute all linear combinations of all sets $S$ containing $L$ with respect to the model outputs $v_{\text{or}}(\mathbf{x}_{N \setminus L})$ given a set of input variables $L$, *i.e.*, $\sum_{S:S \cap T \neq \emptyset, S \supseteq L} (-1)^{|S|-|L|} v_{\text{or}}(\mathbf{x}_{N \setminus L})$. Then, we compute all summations over the set $L \subseteq N$.

In this way, we can compute them separately for different cases of $L \subseteq S \subseteq N, S \cap T \neq \emptyset$. In the following, we consider the cases (1) $L = N \setminus T$, (2) $L = N$, (3) $L \cap T \neq \emptyset, L \neq N$, and (4) $L \cap T = \emptyset, L \neq N \setminus T$, respectively.

(1) When $L = N \setminus T$, the linear combination of all subsets $S$ containing $L$ with respect to the model output $v_{\text{or}}(\mathbf{x}_{N \setminus L})$ is $\sum_{S:S \cap T \neq \emptyset, S \supseteq L} (-1)^{|S|-|L|} v_{\text{or}}(\mathbf{x}_{N \setminus L}) = \sum_{S:S \cap T \neq \emptyset, S \supseteq L} (-1)^{|S|-|L|} v_{\text{or}}(\mathbf{x}_T)$. For all sets $S : S \supseteq L, S \cap T \neq \emptyset$ (then $S \neq N \setminus T, S \neq L$), let us consider the linear combinations of all sets $S$ with number $|S|$ for the model output $v_{\text{or}}(\mathbf{x}_T)$, respectively. Let $|S'| := |S| - |L|, (1 \leq |S'| \leq |T|)$, then there are a total of $C_{|T|}^{|S'|}$ combinations of all sets $S$ of order $|S|$. Thus, given $L$, accumulating the model outputs $v_{\text{or}}(\mathbf{x}_T)$ corresponding to all $S \supseteq L$, then

$\sum_{S:S \cap T \neq \emptyset, S \supseteq L} (-1)^{|S|-|L|} v_{\text{or}}(\mathbf{x}_{N \setminus L}) = v_{\text{or}}(\mathbf{x}_T) \cdot \underbrace{\sum_{|S'|=1}^{|T|} C_{|T|}^{|S'|} (-1)^{|S'|}}_{=-1} = -v_{\text{or}}(\mathbf{x}_T)$.

(2) When $L = N$ (then $S = N$), the linear combination of all subsets $S$ containing $L$ with respect to the model output $v_{\mathrm{or}}(\mathbf{x}_{N\setminus L})$ is $\sum_{S:S\cap T\neq\emptyset,S\supseteq L}(-1)^{|S|-|L|}v_{\mathrm{or}}(\mathbf{x}_{N\setminus L}) = (-1)^{|N|-|N|}v_{\mathrm{or}}(\mathbf{x}_{\emptyset}) = v_{\mathrm{or}}(\mathbf{x}_{\emptyset})$.

(3) When $L \cap T \neq \emptyset, L \neq N$, the linear combination of all subsets $S$ containing $L$ with respect to the model output $v_{\mathrm{or}}(\mathbf{x}_{N\setminus L})$ is $\sum_{S:S\cap T\neq\emptyset,S\supseteq L}(-1)^{|S|-|L|}v_{\mathrm{or}}(\mathbf{x}_{N\setminus L})$. For all sets $S : S \supseteq L, S \cap T \neq \emptyset$, let us consider the linear combinations of all sets $S$ with number $|S|$ for the model output $v_{\mathrm{or}}(\mathbf{x}_T)$, respectively. Let us split $|S| - |L|$ into $|S'|$ and $|S''|$, i.e., $|S| - |L| = |S'| + |S''|$, where $S' = \{i|i \in S, i \notin L, i \in N \setminus T\}$, $S'' = \{i|i \in S, i \notin L, i \in T\}$ (then $0 \leq |S''| \leq |T| - |T \cap L|$) and $S' + S'' + L = S$. In this way, there are a total of $C_{|T|-|T\cap L|}^{|S''|}$ combinations of all sets $S''$ of order $|S''|$. Thus, given $L$, accumulating the model outputs $v_{\mathrm{or}}(\mathbf{x}_{N\setminus L})$ corresponding to all $S \supseteq L$, then $\sum_{S:S\cap T\neq\emptyset,S\supseteq L}(-1)^{|S|-|L|}v_{\mathrm{or}}(\mathbf{x}_{N\setminus L}) = v_{\mathrm{or}}(\mathbf{x}_{N\setminus L}) \cdot \sum_{S'\subseteq N\setminus T\setminus L}\underbrace{\sum_{|S''|=0}^{|T|-|T\cap L|}C_{|T|-|T\cap L|}^{|S''|}(-1)^{|S'|+|S''|}}_{=0} = 0$.

(4) When $L \cap T = \emptyset, L \neq N \setminus T$, the linear combination of all subsets $S$ containing $L$ with respect to the model output $v_{\mathrm{or}}(\mathbf{x}_{N\setminus L})$ is $\sum_{S:S\cap T\neq\emptyset,S\supseteq L}(-1)^{|S|-|L|}v_{\mathrm{or}}(\mathbf{x}_{N\setminus L})$. Similarly, let us split $|S| - |L|$ into $|S'|$ and $|S''|$, i.e., $|S| - |L| = |S'| + |S''|$, where $S' = \{i|i \in S, i \notin L, i \in N \setminus T\}$, $S'' = \{i|i \in S, i \in T\}$ (then $0 \leq |S''| \leq |T|$) and $S' + S'' + L = S$. In this way, there are a total of $C_{|T|}^{|S''|}$ combinations of all sets $S''$ of order $|S''|$. Thus, given $L$, accumulating the model outputs $v_{\mathrm{or}}(\mathbf{x}_{N\setminus L})$ corresponding to all $S \supseteq L$, then $\sum_{S:S\cap T\neq\emptyset,S\supseteq L}(-1)^{|S|-|L|}v_{\mathrm{or}}(\mathbf{x}_{N\setminus L}) = v_{\mathrm{or}}(\mathbf{x}_{N\setminus L}) \cdot \sum_{S'\subseteq N\setminus T\setminus L}\underbrace{\sum_{|S''|=0}^{|T|}C_{|T|}^{|S''|}(-1)^{|S'|+|S''|}}_{=0} = 0$.

Please see the complete derivation of the following formula.

$$
\begin{aligned}
\sum_{S:S\cap T\neq\emptyset}I_{\mathrm{or}}(S|\mathbf{x}_T) &= \sum_{S:S\cap T\neq\emptyset}\left[-\sum_{L\subseteq S}(-1)^{|S|-|L|}v_{\mathrm{or}}(\mathbf{x}_{N\setminus L})\right] \\
&= -\sum_{L\subseteq N}\sum_{S:S\cap T\neq\emptyset,S\supseteq L}(-1)^{|S|-|L|}v_{\mathrm{or}}(\mathbf{x}_{N\setminus L}) \\
&= -\left[\sum_{|S'|=1}^{|T|}C_{|T|}^{|S'|}(-1)^{|S'|}\right]\cdot\underbrace{v_{\mathrm{or}}(\mathbf{x}_T)}_{L=N\setminus T} - \underbrace{v_{\mathrm{or}}(\mathbf{x}_{\emptyset})}_{L=N} \\
&\quad - \sum_{L\cap T\neq\emptyset,L\neq N}\left[\sum_{S'\subseteq N\setminus T\setminus L}\left(\sum_{|S''|=0}^{|T|-|T\cap L|}C_{|T|-|T\cap L|}^{|S''|}(-1)^{|S'|+|S''|}\right)\right]\cdot v_{\mathrm{or}}(\mathbf{x}_{N\setminus L}) \\
&\quad - \sum_{L\cap T=\emptyset,L\neq N\setminus T}\left[\sum_{S'\subseteq N\setminus T\setminus L}\left(\sum_{|S''|=0}^{|T|}C_{|T|}^{|S''|}(-1)^{|S'|+|S''|}\right)\right]\cdot v_{\mathrm{or}}(\mathbf{x}_{N\setminus L}) \\
&= -(-1)\cdot v_{\mathrm{or}}(\mathbf{x}_T) - v_{\mathrm{or}}(\mathbf{x}_{\emptyset}) - \sum_{L\cap T\neq\emptyset,L\neq N}\left[\sum_{S'\subseteq N\setminus T\setminus L}0\right]\cdot v_{\mathrm{or}}(\mathbf{x}_{N\setminus L}) \\
&\quad - \sum_{L\cap T=\emptyset,L\neq N\setminus T}\left[\sum_{S'\subseteq N\setminus T\setminus L}0\right]\cdot v_{\mathrm{or}}(\mathbf{x}_{N\setminus L}) \\
&= v_{\mathrm{or}}(\mathbf{x}_T) - v_{\mathrm{or}}(\mathbf{x}_{\emptyset})
\end{aligned}
\tag{9}
$$

**(3) Universal matching theorem of AND-OR interactions.**

With the universal matching theorem of AND interactions and the universal matching theorem of OR interactions, we can easily get $v(\mathbf{x}_T) = v_{\mathrm{and}}(\mathbf{x}_T) + v_{\mathrm{or}}(\mathbf{x}_T) = \sum_{S\subseteq T}I_{\mathrm{and}}(S|\mathbf{x}_T) + \sum_{S\in\{S:S\cap T\neq\emptyset\}\cup\{\emptyset\}}I_{\mathrm{or}}(S|\mathbf{x}_T)$, thus, we obtain the universal matching theorem of AND-OR interactions.

$\square$

**Theorem 3** (proved by Harsanyi (1963)). *The Shapley value $\phi(i)$ of an input variable $i$ can be explained as a uniform allocation of the AND interactions,* i.e., $\phi(i) = \sum_{S \subseteq N: S \ni i} \frac{1}{|S|} I_{\text{and}}(S|\mathbf{x})$.

## D  PROOF OF THE VARIANCE OF AND AND OR INTERACTIONS

In this section, we prove that the variance of $I_{\text{and}}^{'(i)}(T)$ caused by the Gaussian noises $\epsilon_T^{(i)} \sim \mathcal{N}(0, \sigma^2)$ is $\mathbb{E}_{\epsilon_T \sim \mathcal{N}(0,\sigma^2)}[I_{\text{and}}^{'(i)}(T) - E_{\forall S, \epsilon_S \sim \mathcal{N}(0,\sigma^2)} I_{\text{and}}^{'(i)}(S)]^2 = 2^{|T|}\sigma^2$ in Section 2.3.2.

*Proof.* Given $I_{\text{and}}^{'(i)}(T) = I_{\text{and}}^{(i)}(T) + \sum_{T' \subseteq T}(-1)^{|T|-|T'|}\epsilon_{T'}^{(i)}$, the variance of $I_{\text{and}}^{'(i)}(T)$ is $\text{Var}(I_{\text{and}}^{'(i)}(T)) = \text{Var}(I_{\text{and}}^{(i)}(T) + \sum_{T' \subseteq T}(-1)^{|T|-|T'|}\epsilon_{T'}^{(i)})$. As the AND interaction $I_{\text{and}}^{(i)}(T)$ and the Gaussian noise $\epsilon_T^{(i)}$ are independent of each other, then the variance of $I_{\text{and}}^{'(i)}(T)$ can be decomposed to $\text{Var}(I_{\text{and}}^{'(i)}(T)) = \text{Var}(I_{\text{and}}^{(i)}(T)) + \text{Var}(\sum_{T' \subseteq T}(-1)^{|T|-|T'|}\epsilon_{T'}^{(i)}) = \text{Var}(\sum_{T' \subseteq T}(-1)^{|T|-|T'|}\epsilon_{T'}^{(i)})$, this is because here $I_{\text{and}}^{(i)}(T)$ can be regarded as a constant.

Since each Gaussian noise $\epsilon_T^{(i)} \sim \mathcal{N}(0, \sigma^2), \forall T \subseteq N$ is independent and identically distributed, then the variance is $\text{Var}(I_{\text{and}}^{'(i)}(T)) = \text{Var}(\sum_{T' \subseteq T}(-1)^{|T|-|T'|}\epsilon_{T'}^{(i)}) = \text{Var}(\epsilon_{T_1'}^{(i)}) + \text{Var}(\epsilon_{T_2'}^{(i)}) + \cdots + \text{Var}(\epsilon_{T'_{2^{|T|}}}^{(i)}) = 2^{|T|} \cdot \sigma^2$ (there are a total of $2^{|T|}$ subsets for $T' \subseteq T$). $\square$

## E  THE FAITHFULNESS OF THE SPARSITY AND UNIVERSAL-MATCHING.

This section further demonstrates the faithfulness of the sparsity and universal-matching theorem of AND-OR interactions, both theoretically and experimentally.

Theoretically, the faithfulness of the sparsity and universal-matching theorem of AND-OR interactions means that, given an input sample with $n$ variables, we must prove that (1) a well-trained DNN usually just encodes a small number of salient interactions $\Omega$ (Ren et al., 2024), comparing with all $2^n$ potential combinations of the input variables in a given input sample, *i.e.*, $|\Omega| \ll 2^n$, and that (2) the network output $v(\mathbf{x}_S)$ on all $2^n$ randomly masked samples $\{\mathbf{x}_S | S \subseteq N\}$ can be well matched by a few interactions in $\Omega = \{S \subseteq N : |I(S|x)| > \tau\}$ as defined in the Definition of interaction primitives in Section 2.1. These two terms have been proved in Theorem 1, Theorem 2 and Proposition 1.

Then, in practice, considering the $2^n$ computational complexity, we have followed settings in (Li & Zhang, 2023b) to extract interactions between a set of randomly selected input variables $N = \{1, 2, \ldots, t\}, (t < n)$, while other unselected $(n - t)$ input variables remain unmasked, leaving the original state unchanged. In this case, faithfulness does not mean that our interactions explain all the inference logic encoded between all $n$ input variables for a given sample $x$ in the pre-trained DNN. Instead, it only means that the extracted interactions can also accurately match the inference logic encoded between the selected $t$ input variables for a given sample $x$ in the DNN.

**Experimental verification.** In addition, we have further conducted an experiment to verify the faithfulness when we explain interactions between all input variables. To this end, for the sentiment classification task on the SST-2 dataset in BERT$_{\text{BASE}}$, we selected sentences containing 15 tokens to verify the faithfulness of the sparsity in Proposition 1 and the universal-matching property in Theorem 2. All tokens are selected as input variables. We focused on the matching errors for all masked samples $\mathbf{x}_T, \forall T \subseteq N$. Specifically, we observed whether the real network output on the masked sample $v(\mathbf{x}_T), \forall T \subseteq N$ can be well approximated by interactions. We have verified that the extracted salient interactions in $\Omega$ faithfully explain the network output, *i.e.*, $\forall T \subseteq N, v(\mathbf{x}_T) \approx v(\mathbf{x}_\emptyset) + \sum_{\emptyset \neq S \subseteq T: S \in \Omega^{\text{and}}} I_{\text{and}}(S|\mathbf{x}_T) + \sum_{S \cap T \neq \emptyset: S \in \Omega^{\text{or}}} I_{\text{or}}(S|\mathbf{x}_T)$. Figure 6 illustrates that network output $v(\mathbf{x}_T), \forall T \subseteq N$ on all $2^n$ randomly masked samples can be well fitted by interactions.

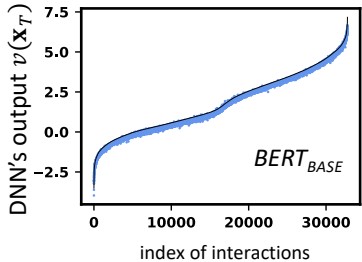

Figure 6: Universal matching of all interaction primitives to the DNN's output, when we use salient interactions to match the DNN's output. Shade area indicates the matching error of different $v(\mathbf{x}_T)$.

## F  DETAILED EXPLANATION OF EQUATION (6)

This section uses a simple example to illustrate Equation (6) more clearly. Let us illustrate how the interaction matrices are formatted in a toy example. Let us consider two pre-trained DNNs $v^{(1)}$, $v^{(2)}$ and an input sample $\mathbf{x}$ with $N = \{1, 2\}$ variables, and take the AND interaction matrix $\mathbb{I}_{\text{and}}$ in Equation (6) as an example (the OR interaction matrix $\mathbb{I}_{\text{or}}$ can be obtained similarly). First, we get all $2^2$ AND interactions extracted from the $i$-th model $\mathbf{I}_{\text{and}}^{(i)} = [I_{\text{and}}(T_1|\mathbf{x}), I_{\text{and}}(T_2|\mathbf{x}), I_{\text{and}}(T_3|\mathbf{x}), I_{\text{and}}(T_4|\mathbf{x})]^{\mathsf{T}} \in \mathbb{R}^{2^2}$, according to the description of Equation (4). Here, each interaction value $I_{\text{and}}(T_k|\mathbf{x}) \in \mathbb{R}, T_k \subseteq N$ denotes the interaction value of each masked sample $\mathbf{x}_{T_k}$, which is computed according to Equation (1). Second, the AND interaction matrix $\mathbb{I}_{\text{and}} = \left[\mathbf{I}_{\text{and}}^{(1)}, \mathbf{I}_{\text{and}}^{(2)}\right] \in \mathbb{R}^{2^2 \times 2}$ represents the interaction values corresponding to the $2^2$ masked samples for each of the two models.

Let us illustrate how to learn the parameter $\gamma_T^{(i)}$ in Equation (6). The loss in Equation (6) in the above example can be represented as the function of $\{\gamma_T\}$, as follows.

$$
\begin{aligned}
Loss &= \min_{\{\gamma_{T_k}^{(1)}, \gamma_{T_k}^{(2)}\}} \left( \|\text{rowmax}(\mathbb{I}_{\text{and}})\|_1 + \|\text{rowmax}(\mathbb{I}_{\text{or}})\|_1 \right) + \alpha \left( \|\mathbb{I}_{\text{and}}\|_1 + \|\mathbb{I}_{\text{or}}\|_1 \right) \\
&= \min_{\{\gamma_{T_k}^{(1)}, \gamma_{T_k}^{(2)}\}} \sum_{T_k \subseteq N} |\max(\sum_{T \subseteq S}(-1)^{|S|-|T|}[0.5 \cdot v^{(1)}(\mathbf{x}_{T_k}) + \gamma_{T_k}^{(1)}], \sum_{T \subseteq S}(-1)^{|S|-|T|}[0.5 \cdot v^{(2)}(\mathbf{x}_{T_k}) + \gamma_{T_k}^{(2)}])| \\
&\quad + \sum_{T_k \subseteq N} |\max(-\sum_{T \subseteq S}(-1)^{|S|-|T|}[0.5 \cdot v^{(1)}(\mathbf{x}_{N \setminus T_k}) - \gamma_{T_k}^{(1)}], -\sum_{T \subseteq S}(-1)^{|S|-|T|}[0.5 \cdot v^{(2)}(\mathbf{x}_{N \setminus T_k}) - \gamma_{T_k}^{(2)}])| \\
&\quad + \alpha \cdot \sum_{T_k \subseteq N} \sum_{i=1}^{2} |\sum_{T \subseteq S}(-1)^{|S|-|T|}[0.5 \cdot v^{(i)}(\mathbf{x}_{T_k}) + \gamma_{T_k}^{(i)}]| \\
&\quad + \alpha \cdot \sum_{T_k \subseteq N} \sum_{i=1}^{2} |-\sum_{T \subseteq S}(-1)^{|S|-|T|}[0.5 \cdot v^{(i)}(\mathbf{x}_{N \setminus T_k}) - \gamma_{T_k}^{(i)}])|.
\end{aligned}
\tag{10}
$$

Therefore, we only need to optimize $\{\gamma_{T_k}^{(1)}, \gamma_{T_k}^{(2)}\}$ via gradient descent to reduce the loss in Equation (6).

## G  THE RUN-TIME COMPLEXITY

This section explores the run-time complexity of extracting AND-OR interactions on different tasks. Theoretically, given an input sample with $n$ input variables, the time complexity is $2^n$, and we need to generate masked samples for model inference. Fortunately, the variable number is not too large by following the settings in (Li & Zhang, 2023b) and (Shen et al., 2023). In real applications,

Table 1: Average run-time per sample on different tasks.

| sentiment classification on the SST-2 dataset *w.r.t.* the pair of models (BERT$_{\text{BASE}}$ and BERT$_{\text{LARGE}}$) | dialogue task on the SQuAD dataset *w.r.t.* the pair of models (LLaMA and OPT-1.3B) | image classification on the MNIST dataset *w.r.t.* the pair of models (ResNet-20 and VGG-16) |
|---|---|---|
| 45.14 seconds | 46.61 seconds | 27.15 seconds |

the average running time for a sentence in the SST-2 dataset on the BERT$_{\text{LARGE}}$ model is 45.14 seconds. Table 1 further shows the average running time of each input sample for different tasks.

# H MORE BASELINE TO VERIFY THE EFFECTIVENESS OF EQUATION (6)

To further verify the effectiveness of the interactions extracted from Equation (6), we conducted experiments on more baseline, namely the original Harsanyi interaction (Ren et al., 2023a). We compared the sparsity and the generalization power of the AND interactions extracted from different DNNs. Figure 7 shows that the interactions extracted by our method exhibit higher sparsity and generalization power compared to the original AND interactions.

To enable fair comparisons, we double the number of Harsanyi interactions extracted from a sample by setting $I'(S|x) = I(S|x)$. Thus, we congregate both sets of interactions (a total of $2 \cdot 2^n$ interactions) to draw a curve in Figure 2 and Figure 3. In this way, we can compare the same number of interactions over different methods.

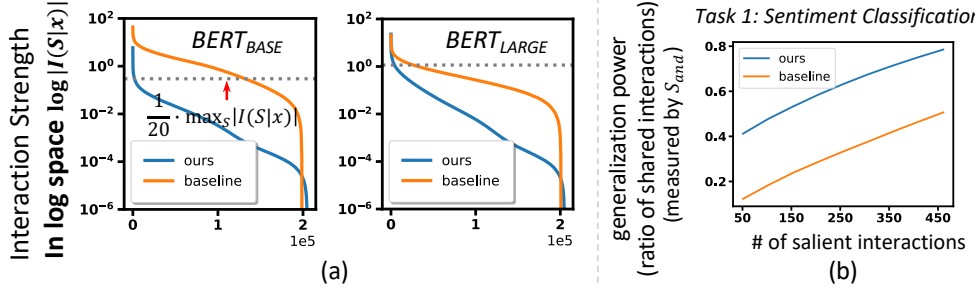

Figure 7: Comparing the sparsity and generalization power of the extracted interactions between the original Harsanyi interactions and our proposed method.

# I ABLATION STUDY OF THE PARAMETER $\alpha$ IN EQUATION (6)

To explore the effect of $\alpha$ on the AND-OR interactions extracted from Equation (6), we conducted an ablation study on $\alpha$. Specifically, we jointly extracted two sets of AND-OR interactions from the BERT$_{\text{BASE}}$ and BERT$_{\text{LARGE}}$ models, which were trained by (Devlin et al., 2019) and further finetuned by us for sentiment classification on the SST-2 dataset. For comparison, we set the value of $\alpha$ to [0, 0.2, 0.4, 0.6, 0.8, 1.0], respectively. Here, when $\alpha = 0$, Equation (6) degenerated into Equation (5), indicating that only the largest interactions among the $m$ DNNs were penalized. As $\alpha$ increases, the effects of not-so-salient interactions in other models were taken into account. Figure 8 shows that as the value of $\alpha$ increases, the sparsity of the extracted interactions did not increase too much, but the generalization power of the extracted interactions decreased. This shows the ef-

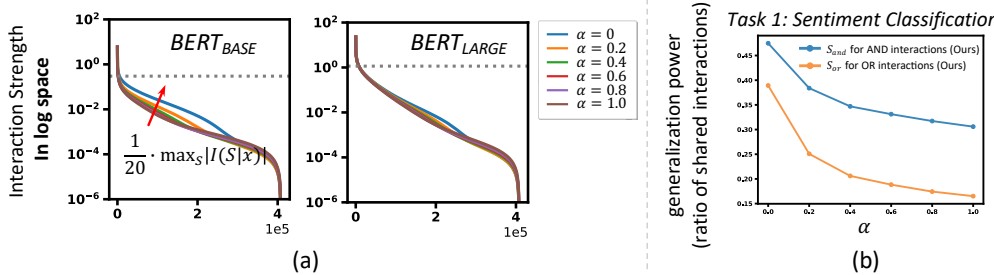

Figure 8: Effect of $\alpha$ on the AND-OR interactions extracted from Equation (6). (a) Strength of all interactions extracted from all samples, which were sorted in a descending order. The increase of different $\alpha$ value did not significantly affect the sparsity of interactions. (b) Decreasing generalization power of extracted AND-OR interactions, when the $\alpha$ value increased.

fectiveness of the penalty in Equation (5) in boosting the generalization power without significantly hurting the sparsity.

## J VISUALIZATION OF MORE SHARED AND DISTINCTIVE INTERACTION PRIMITIVES FOR FIGURE 5

In this section, we visualized all shared and distinctive interaction primitives across different DNNs in Figure 5. Specifically, we randomly selected 10 tokens in the given sentence, labeling these 10 tokens in red and the other unselected tokens in black. Here, $n = 10$ input variables denote the embeddings corresponding to these 10 tokens. Since each input variable has two states, masked and unmasked, a total of $2^{10}$ masked samples are generated. Then, according to Equations (1) and (2), a total of $2 \cdot 2^{10}$ AND interactions and OR interactions are finally obtained.

Then, we extracted the most salient $k = 50$ interactions from a total of $2 \cdot 2^{10}$ AND interactions and OR interactions as the set of AND-OR interaction primitives in each DNN, respectively. Therefore, in our proposed method, 25 shared interactions were extracted from both models, and 25 distinctive interactions were extracted from the BERT$_{BASE}$ and BERT$_{LARGE}$ models, respectively. In contrast, in the traditional method, 16 shared interactions were extracted from both models, and 34 distinctive interactions were extracted from the BERT$_{BASE}$ and BERT$_{LARGE}$ models, respectively.

In addition, Figure 9 shows the strength of the interaction value $|I(S|\mathbf{x})|$ for each salient interaction. Specifically, we show the strength of the interaction value for each distinctive interaction in each DNN. We also show the strengths of two interaction values for each shared interaction extracted from both DNNs, where the strength of the interaction value on the BERT$_{BASE}$ model is on the left and the strength of the interaction value on the BERT$_{LARGE}$ model is on the right. Figure 9 illustrates that, our method extracted more shared interactions compared to the traditional interaction-extraction method.

## K DISCUSSION ON THE DIFFERENCES IN THE SIMILARITY OF INTERACTIONS

This section compares the consistency of interactions between different types of DNNs. Since LLaMA (Touvron et al., 2023) and OPT-1.3B (Zhang et al., 2022b) have much more parameters than BERT$_{BASE}$ and BERT$_{LARGE}$ (Devlin et al., 2019), it is often widely believed that these two LLMs can better converge to the true language knowledge in the training data.

We can roughly understand this phenomenon by clarifying the following two phenomena. First, through extensive experiments, the authors have found that the learning of a DNN usually has two phases, *i.e.*, the learning of new interactions and the forgotten of incorrectly learned interactions (this is our on-going research). In most cases, after the forgetting phase, the remained interactions of an LLM are usually shared by other LLMs. The high similarity of interactions between LLMs has also been observed in (Shen et al., 2023).

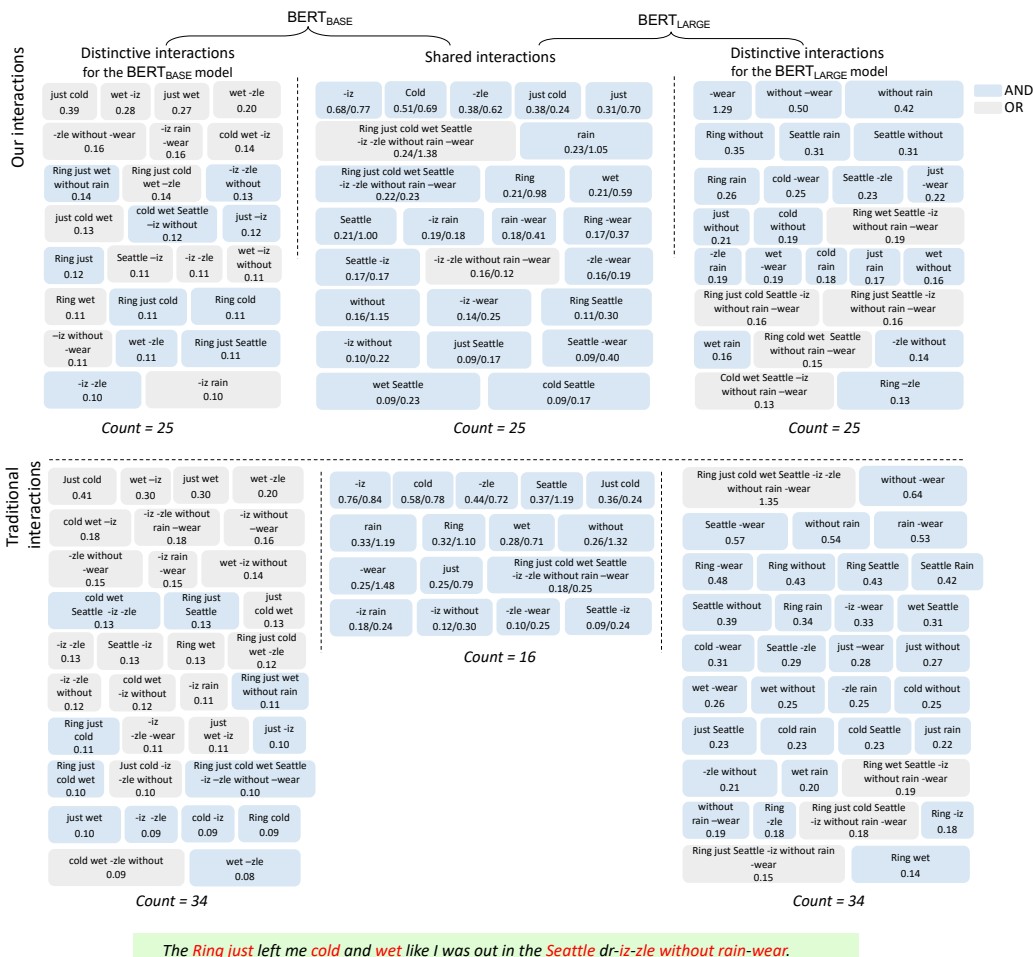

Figure 9: Visualization of more shared and distinctive interaction primitives for Figure 5.

Second, compared to LLMs, relatively small models are less powerful to remove all incorrectly learned interactions. For example, a simple model is usually less powerful to regress a complex function. The less powerful models (BERT$_{\text{BASE}}$ and BERT$_{\text{LARGE}}$) are not powerful enough to accurately encode the potentially complex interactions.

Therefore, small models are more likely to represent various incorrect interactions to approximate the true target of the task. This may partially explains the high difficulty of extracting common interactions from two BERT models, as well as why the proposed method shows more improvements in the generalization power of interactions on BERT models.

**Experimental verification.** To this end, we have further conducted a new experiment to compare the interactions extracted from two LLMs, *i.e.*, LLaMA (Touvron et al., 2023) and Aquila-7B (BAAI, 2023). As Figure 10 shows, two LLMs usually encode much more similar interactions than two not-so-large models. In fact, we have also observed similar interactions in anther pair of LLMs, *i.e.*, LLaMA and OPT-1.3B (Zhang et al., 2022b) (please see Figure 4). This partially explains the reason why the performance improvement of these two LLMs is similar to that of LLaMA and OPT-1.3B.

## L  A CONCRETE EXAMPLE TO ILLUSTRATE THE PROCEDURE

Let us use a concrete input sentence with six tokens to illustrate the experimental procedure from beginning to end.

**Step 1:** Given the sentence "A stitch in time saves nine" with six tokens, the six input variables are $x_1$ = embedding of token "A", $x_2$ = embedding of token "stitch", $x_3$ = embedding of token "in",

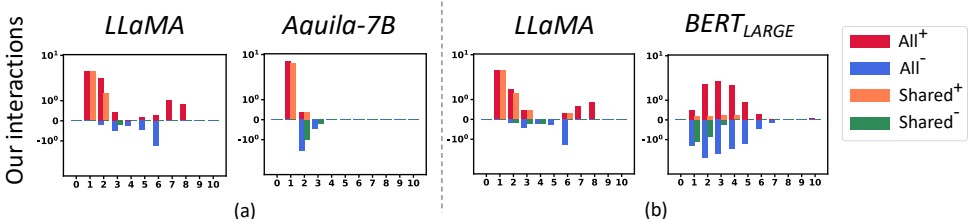

Figure 10: Differences of interactions between different types of DNNs.

$x_4$ = embedding of token "time", $x_5$ = embedding of token "saves", and $x_6$ = embedding of token "nine". Although the dimensions of the embedding for the BERT$_\text{BASE}$ and BERT$_\text{LARGE}$ models are different, we can still compute the interactions between the embeddings corresponding to the same token. Specifically, the embedding $x_i$ of a token in the BERT$_\text{BASE}$ model is $x_i \in \mathbb{R}^{768}$, and the embedding $x_i$ of a token in the BERT$_\text{LARGE}$ model is $x_i \in \mathbb{R}^{1024}$.

**Step 2:** The baseline value for each input variable is $b_1 = b_2 = b_3 = b_4 = b_5 = b_6$ = special embedding of a masked token, where the special embedding of the masked token is encoded by the BERT$_\text{BASE}$ and BERT$_\text{LARGE}$ model, respectively. This special embedding can use the embedding of a special token, *e.g.*, the embedding of the [CLS] token. Specifically, the baseline $b_i$ in the BERT$_\text{BASE}$ model is $b_i \in \mathbb{R}^{768}$, and the baseline $b_i$ in the BERT$_\text{LARGE}$ model is $b_i \in \mathbb{R}^{1024}$.

**Step 3:** In this case, there are a total of $2^6$ masked samples $\mathbf{x}_{T_0}, \mathbf{x}_{T_1}, \mathbf{x}_{T_2}, \mathbf{x}_{T_3}, \cdots, \mathbf{x}_{T_{62}}, \mathbf{x}_{T_{63}}$ used for the model inference. Specifically, the first masked sample is $\mathbf{x}_{T_0} = \{b_1, b_2, b_3, b_4, b_5, b_6\}$, where each of its input variables (embedding) is replaced with the corresponding baseline value (embedding of a special token), *i.e.*, $x_i = b_i, i \in \{1, 2, \cdots, 6\}$. The second masked sample is $\mathbf{x}_{T_1} = \{b_1, b_2, b_3, b_4, b_5, x_6\}$, where its input variable $x_6$ is kept as the original embedding, and the other five input variables are replaced with the corresponding baseline values, *i.e.*, $x_1 = b_1, x_2 = b_2, x_3 = b_3, x_4 = b_4, x_5 = b_5$. The third masked sample is $\mathbf{x}_{T_2} = \{b_1, b_2, b_3, b_4, x_5, b_6\}$, where its input variable $x_5$ is kept as the original embedding, and the other five input variables are replaced with the corresponding baseline values, *i.e.*, $x_1 = b_1, x_2 = b_2, x_3 = b_3, x_4 = b_4, x_6 = b_6$. The fourth masked sample is $\mathbf{x}_{T_3} = \{b_1, b_2, b_3, b_4, x_5, x_6\}$, where its input variables $x_5$ and $x_6$ are kept as the original embedding, respectively, and the other four input variables are replaced with the corresponding baseline values. Similarly, the 63rd masked sample is $\mathbf{x}_{T_{62}} = \{x_1, x_2, x_3, x_4, x_5, b_6\}$, where its input variables $x_1$, $x_2$, $x_3$, $x_4$, $x_5$ are kept as the original embedding, respectively, and $x_6 = b_6$. The 64th masked sample is $\mathbf{x}_{T_{63}} = \{x_1, x_2, x_3, x_4, x_5, x_6\}$, where all input variables are kept unchanged from the original embedding.

**Step 4:** For each masked sample $\mathbf{x}_{T_j}, j \in \{0, 1, \cdots, 63\}$, we computed the log-odds output of the ground-truth label $v(\mathbf{x}_{T_j}) = \log \frac{p(y=y^{\text{truth}}|\mathbf{x}_{T_j})}{1-p(y=y^{\text{truth}}|\mathbf{x}_{T_j})} \in \mathbb{R}$ as the model output. In this way, feeding all masked samples $\mathbf{x}_{T_j}$ into the BERT$_\text{BASE}$ model produced a total of $2^6$ model outputs $v^{\text{BASE}}(\mathbf{x}_{T_0})$, $v^{\text{BASE}}(\mathbf{x}_{T_1}), \cdots, v^{\text{BASE}}(\mathbf{x}_{T_{63}})$. Feeding all masked samples $\mathbf{x}_{T_j}$ into the BERT$_\text{LARGE}$ model produced a total of $2^6$ model outputs $v^{\text{LARGE}}(\mathbf{x}_{T_0}), v^{\text{LARGE}}(\mathbf{x}_{T_1}), \cdots, v^{\text{LARGE}}(\mathbf{x}_{T_{63}})$.

**Step 5:** Computed the AND outputs $v_{\text{and}}^{\text{BASE}}(\mathbf{x}_{T_j}) = 0.5v^{\text{BASE}}(\mathbf{x}_{T_j}) + \gamma_{T_j}^{\text{BASE}}, j \in \{0, 1, \cdots, 63\}$ and the OR outputs $v_{\text{or}}^{\text{BASE}}(\mathbf{x}_{T_j}) = 0.5v^{\text{BASE}}(\mathbf{x}_{T_j}) - \gamma_{T_j}^{\text{BASE}}, j \in \{0, 1, \cdots, 63\}$ for the BERT$_\text{BASE}$ model, respectively. Computed the AND outputs $v_{\text{and}}^{\text{LARGE}}(\mathbf{x}_{T_j}) = 0.5v^{\text{LARGE}}(\mathbf{x}_{T_j}) + \gamma_{T_j}^{\text{LARGE}}, j \in \{0, 1, \cdots, 63\}$ and the OR outputs $v_{\text{or}}^{\text{LARGE}}(\mathbf{x}_{T_j}) = 0.5v^{\text{LARGE}}(\mathbf{x}_{T_j}) - \gamma_{T_j}^{\text{LARGE}}, j \in \{0, 1, \cdots, 63\}$ for the BERT$_\text{LARGE}$ model, respectively.

**Step 6:** Computed the AND interactions $I_{\text{and}}^{\text{BASE}}(T_j|\mathbf{x}) = \sum_{T \subseteq T_j}(-1)^{|T_j|-|T|}v_{\text{and}}^{\text{BASE}}(\mathbf{x}_T), j \in \{0, 1, \cdots, 63\}$ and the OR interactions $I_{\text{or}}^{\text{BASE}}(T_j|\mathbf{x}) = -\sum_{T \subseteq T_j}(-1)^{|T_j|-|T|}v_{\text{or}}^{\text{BASE}}(\mathbf{x}_{N \setminus T}), j \in \{0, 1, \cdots, 63\}$ for the BERT$_\text{BASE}$ model, respectively. Computed the AND interactions $I_{\text{and}}^{\text{LARGE}}(T_j|\mathbf{x}) = \sum_{T \subseteq T_j}(-1)^{|T_j|-|T|}v_{\text{and}}^{\text{LARGE}}(\mathbf{x}_T), j \in \{0, 1, \cdots, 63\}$ and the OR interactions $I_{\text{or}}^{\text{LARGE}}(T_j|\mathbf{x}) = -\sum_{T \subseteq T_j}(-1)^{|T_j|-|T|}v_{\text{or}}^{\text{LARGE}}(\mathbf{x}_{N \setminus T}), j \in \{0, 1, \cdots, 63\}$ for the BERT$_\text{LARGE}$ model, respectively.

Table 2: Diversity of two sets of interactions, which are extracted based on Equation (4) from the same models but with different initialized parameters $\{\gamma_T\}$.

| DNNs | $\mathcal{S}_{\text{and}}$ | $\mathcal{S}_{\text{or}}$ |
|---|---|---|
| $\text{BERT}_{\text{BASE}}$ | 10.90% | 16.18% |
| $\text{BERT}_{\text{LARGE}}$ | 17.84% | 20.90% |

**Step 7:** Learned the parameters $\gamma_{T_j}^{\text{BASE}}$ and $\gamma_{T_j}^{\text{LARGE}}, j \in \{0, 1, \cdots, 63\}$ using the loss in Equation (6). Went back to Step 5 and repeated the iterations until the loss converges.

**Step 8:** Computed the final AND interactions $I_{\text{and}}^{\text{BASE}}(T_j|\mathbf{x})$ and the final OR interactions $I_{\text{or}}^{\text{BASE}}(T_j|\mathbf{x}), j \in \{0, 1, \cdots, 63\}$ for the $\text{BERT}_{\text{BASE}}$ model using the learnable parameters $\gamma_{T_j}^{\text{BASE}}, j \in \{0, 1, \cdots, 63\}$. The final salient interactions of the $\text{BERT}_{\text{BASE}}$ model are obtained from all $2^6$ interactions, where the final salient interactions are those with interaction values greater than the threshold $\tau^{\text{BASE}}$, $\Omega^{\text{BASE}} = \{T_j \subseteq N : |I_{\text{and/or}}^{\text{BASE}}(T_j|\mathbf{x})| > \tau^{\text{BASE}}\}$.

Computed the final AND interactions $I_{\text{and}}^{\text{LARGE}}(T_j|\mathbf{x})$ and the final OR interactions $I_{\text{or}}^{\text{LARGE}}(T_j|\mathbf{x}), j \in \{0, 1, \cdots, 63\}$ for the $\text{BERT}_{\text{LARGE}}$ model using the learnable parameters $\gamma_{T_j}^{\text{LARGE}}, j \in \{0, 1, \cdots, 63\}$. The final salient interactions of the $\text{BERT}_{\text{LARGE}}$ model are obtained from all $2^6$ interactions, where the final salient interactions are those with interaction values greater than the threshold $\tau^{\text{LARGE}}$, $\Omega^{\text{LARGE}} = \{T_j \subseteq N : |I_{\text{and/or}}^{\text{LARGE}}(T_j|\mathbf{x})| > \tau^{\text{LARGE}}\}$.

## M  MATCHING PRECISION OF AND-OR INTERACTIONS

In this section, we conducted experiments to show the matching precision of the interactions used for matching the network output. Specifically, we extracted traditional AND-OR interaction primitives using various DNNs trained on different datasets following the settings in (Li & Zhang, 2023b). Then, we used the metric $m = \frac{\sum_{S \in \{\text{top } k \text{ interactions}\}} |I(S)|}{\sum_{S \in \{\text{top } k \text{ interactions}\}} |I(S)| + |v(N) - v(\emptyset) - \sum_{S \in \{\text{top } k \text{ interactions}\}} I(S)|}$ to measure the matching precision of the salient interactions. Figure 11 shows the matching precision of the interactions when we computed $m$ based on different numbers $k$ of most salient interactions. We found that only a few salient interactions were required to achieve a relatively high matching precision.

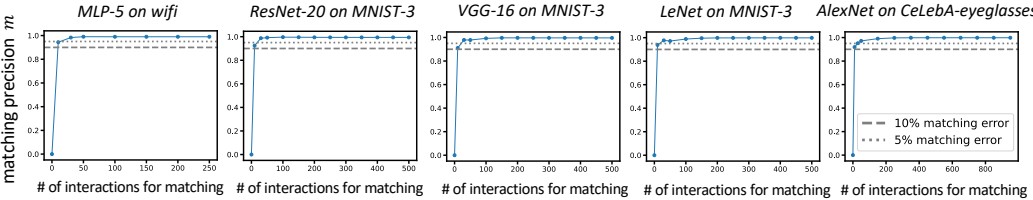

Figure 11: Matching precision of the interactions used for matching the network output.

## N  MORE EXPERIMENTS

### N.1  OPTIMIZING THE LOSS IN EQUATION (4) MAY LEAD TO DIVERSE SOLUTIONS

In Section 2.3.1, we mentioned that optimizing the loss in Equation (4) may lead to diverse solutions. Given different initial states, optimizing the loss in Equation (4) usually extracted two different sets of AND-OR interactions. In particular, as shown in Table 2, when we set the initial state of two sets of parameters $\{\gamma_T\}$ as $\gamma_T \sim \mathcal{N}(0, 1)$, the loss in Equation (4) would learned dramatically different interactions with only 21% overlap. Figure 12 further shows the top 5 AND-OR interaction primitives extracted from $\text{BERT}_{\text{LARGE}}$ model on an input sentence. Both experiments illustrate that given different initial states, the loss in Equation (4) may learn different AND-OR interactions.

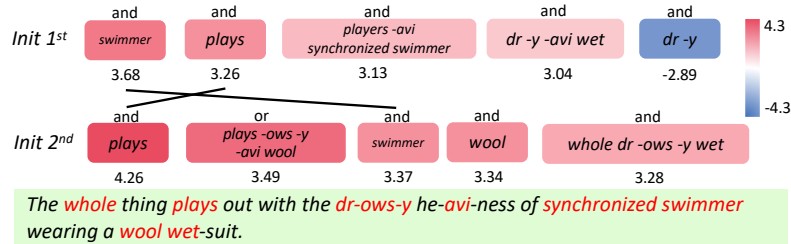

Figure 12: Top 5 AND-OR interaction primitives extracted from BERT$_{\text{LARGE}}$ model, when optimizing the loss in Equation (4) given different initial states.

### N.2 Examining whether the interactions can explain the network output

In Section 2.3.2, we mentioned to conduct experiments to examine whether the extracted AND-OR interactions could still accurately explain the network output, when we removed the error term. Figure 13 shows matching errors of all masked samples for all subsets $T \subseteq N$, when we sorted the network outputs for all $2^n$ masked samples in a descending order. It shows that the real network output was well approximated by interactions.

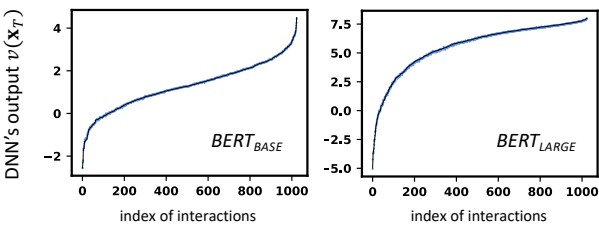

Figure 13: Universal matching of interaction primitives to the DNN's output. Shade area indicates the matching error of different $v(\mathbf{x}_T)$.

## O  Experimental details

### O.1 Performance of DNNs

In Section 3, we conducted experiments using several DNNs trained on different types of datasets, including the language and image datasets. In the sentiment classification task, we fintuned the pretrained models, BERT$_{\text{BASE}}$ and BERT$_{\text{LARGE}}$, using the SST-2 dataset. For image classification task, we trained ResNet-20 and VGG-16 with the MNIST-3 dataset. Table 3 reports the classification accuracy of the aforementioned DNNs. For the dialogue task, we used the pretrained models, the LLaMA model and OPT-1.3B model, directly.

Table 3: Classification accuracy of different DNNs in the sentiment classification and image classification tasks.

| Tasks | dataset | DNNs | |
|-------|---------|------|---|
| sentiment classification | *SST-2* | BERT$_{\text{BASE}}$ 91.32% | BERT$_{\text{LARGE}}$ 93.26% |
| image classification | *MNIST-3* | ResNet-20 100% | VGG-16 100% |

### O.2 The selection of input variables for interaction extraction

This section discusses the selection of input variables for extracting interactions. As mentioned in Section 2.2, given an input sample $\mathbf{x}$ with $n$ input variables, we can extracted at most $2^n$ interac-

tions. Therefore, the computational cost for extracting interactions increases exponentially with the number of input variables. For example, if we take a word in a sentence (or a pixel in an image) as an input variable, the computation is usually inapplicable. To alleviate this issue, we followed (Shen et al., 2023) to select a set of words as input variables and leave other words as the constant background to compute interactions between them. Specifically, we selected 8-10 input variables for each sample in the three tasks. We only extracted the interactions between the selected variables, leaving the rest of the unselected input variables unchanged as background.

• For sentences in the SST-2 dataset, we first tokenized the input sentences and selected tokens as input variables for 200 samples. For each sentence in this dataset, some words have no clear semantics for sentiment classification, *i.e.*, stop words containing dummy words and pronouns, and consequently, there is little interaction within their corresponding tokens. Therefore, we bypassed these tokens without clear semantics, and only selected tokens among the remaining semantic tokens. To facilitate analysis, we randomly selected $n = 10$ tokens as input variables for sentences with more than 10 semantic tokens. The masking of input variables was performed at the embedding level.

• For sentences in the SQuAD dataset, we took first several words in each document as the input sample to the DNN. Specifically, we selected the first 30 words in each document as the input sample and the 31st word as the target word, provided that the following conditions were met: 1) The 31st word possessed clear semantic meaning, which means that it did not belong to the category of stop words. 2) The five words immediately preceding the target word did not constitute sentence-ending punctuation marks, such as a full stop. If either of these conditions was not satisfied, we extended the initial 30 words until all requirements were fulfilled, and selected the next word as the target word. When extracting interactions, we randomly selected a set of $n = 10$ words which have semantic meanings as input variables. It's worth noting that a single word can correspond to multiple tokens, so when we masked a specific word, we masked all of its corresponding tokens. The masking of input variables was performed at the embedding level.

• For images in the MNIST-3 dataset, we manually labeled the semantic part for 100 positive samples (digit 3) by following (Li & Zhang, 2023b). For each image in this dataset, most of the pixels are black background pixels with no semantic information, and consequently, there is no interaction within these black pixels. Thus, we considered interactions only within foreground pixels. Specifically, we divided a whole image into small patches of size $3 \times 3$ and selected $n = 8$ patches in the foreground of an image as input variables. Following (Li & Zhang, 2023b), we used the zero patches as the baseline values to mask the unselected patches $i \in N \setminus T$ in the sample.

