# OpenReview forum: "Defining and extracting generalizable interaction primitives from DNNs"
_ICLR.cc/2024/Conference — ICLR 2024 poster_

### Official Review · Reviewer_MRpB · 2023-10-31

**Soundness:** 3 good
**Presentation:** 3 good
**Contribution:** 4 excellent
**Rating:** 8
**Confidence:** 4

**Summary:**

This paper delves into the realm of deep neural networks (DNNs), addressing a critical gap in understanding the interactions encoded by these models. Unlike existing interaction extraction methods, which lack theoretical guarantees for transferability, this work introduces a precise definition for the transferability of interaction primitives. Building on this foundation, the authors propose a novel method aimed at extracting interactions with maximal generalization power across different DNN architectures. Extensive experiments across language and image domains validate the effectiveness of the proposed method, demonstrating enhanced generalization of extracted interactions.

This paper makes a significant stride in demystifying the interactions encoded by deep neural networks, introducing novel concepts and methodologies backed by solid experimental evidence. Addressing the highlighted weaknesses and questions will further fortify the paper's contributions, making it a valuable addition to the field of DNN interpretability.

**Strengths:**

1. Relevance and Timeliness: Given the burgeoning interest in interpretability of DNNs, this paper tackles a pertinent and significant problem, contributing to a better understanding of how DNNs encode interactions.

2. Innovative Conceptualization: The introduction of "interaction primitive transferability" is a novel and insightful contribution, offering a new lens through which to evaluate and understand interaction primitives in DNNs.

3. Rigorous Experimental Validation: The authors have conducted a comprehensive set of experiments across various settings and domains, including both language and image data, which robustly demonstrate the effectiveness of the proposed method.

**Weaknesses:**

1. Clarification on Parameter Selection: The paper could benefit from a more detailed discussion on the choice of the parameter α, including its impact on the balance between two competing objectives and its sensitivity to variations.

2. Inconsistencies in Performance Improvements: The observed discrepancies in performance improvements between different tasks (i.e., more pronounced improvements in BERT-base and BERT-large compared to LLaMA and OPT-1.3B) warrant a deeper investigation and explanation.

3. Interpretability of Extracted Interactions: While the paper demonstrates the effectiveness of the proposed method in extracting distinctive interactions, a more explicit exploration of the interpretability of these interactions, especially in the context of more powerful models like BERT-large, would add valuable insights.

**Questions:**

1. Parameter Sensitivity: Could the authors provide a more comprehensive analysis on the choice and sensitivity of α? An ablation study exploring how variations in α affect the outcomes would enhance the paper’s rigor.

2. Discrepancies in Performance: What factors contribute to the observed variations in performance improvements across different tasks? A detailed examination of this phenomenon would provide clarity and strengthen the paper’s contributions.

3. Exploration of Interpretability: Can the authors elaborate on whether the distinctive interactions extracted from more powerful models, such as BERT-large, result in enhanced interpretability? An in-depth discussion on this aspect would be highly beneficial.

---

> ### Author Response · Authors · 2023-11-18
>
> Thank you for your insightful comments and suggestions. We would like to answer all the questions and clarify your concerns. We have carefully revised the paper in blue according to your suggestions.
>
> **Q1: Asking to conduct ablation studies.**
>
> > Q1: "a more detailed discussion on the choice of the parameter $\alpha$" "An ablation study exploring how variations in $\alpha$ affect the outcomes"
>
> **A1:** Thank you. We have followed your suggestions to **conduct a new ablation study** to illustrate the impact of $\alpha$. Specifically, we jointly extracted two sets of AND-OR interactions from the $\rm BERT_{BASE}$ and $\rm BERT_{LARGE}$ models, which were trained by [cite 1] and further finetuned by us for sentiment classification on the SST-2 dataset. We set the $\alpha$ values to [0, 0.2, 0.4, 0.6, 0.8, 1.0] for comparison. Figure 8 in the revised paper shows that as $\alpha$ increases, the sparsity of the extracted interactions increased and the generalization power of the extracted interactions gradually decreased. Please see Figure 8 in Appendix K for details of the new ablation study.
>
> [cite 1] Devlin, J., et al. Bert: Pre-training of deep bidirectional transformers for language understanding. In NAACL, 2019.
>
>
>
> **Q2: Inconsistencies in performance Improvements.**
>
> > Q2: "The observed discrepancies in performance improvements … warrant a deeper investigation and explanation."
>
> **A2:** Thank you. This is a deep insight into the LLMs in feature representations. Since LLaMA and OPT-1.3B have much more parameters than BERT-Base and BERT-Large, it is often widely believed that these two LLMs can better converge to the true language knowledge in the training data. Although the following answer is far beyond the research in this paper, we are still glad to answer this question.
>
> We can roughly understand this phenomenon by clarifying the following two phenomena. **First, through extensive experiments, the authors have found that the learning of a DNN usually has two phases, i.e., the learning of new interactions and the forgotten of incorrectly learned interactions** (this is our on-going research). In most cases, after the forgetting phase, the remained interactions of an LLM are usually shared by other LLMs. The high similarity of interactions between LLMs has also been observed in [cite 2].
>
> **Second, compared to LLMs, relatively small models are less powerful to remove all incorrectly learned interactions.** For example, a simple model is usually less powerful to regress a complex function. The less powerful models (BERT-Base and BERT-Large) are not powerful enough to accurately encode the potentially complex interactions.
>
> **Therefore, small models are more likely to represent various incorrect interactions to approximate the true target of the task. This may partially explains the high difficulty of extracting common interactions from two BERT models, as well as why the proposed method shows more improvements in the generalization power of interactions on BERT models.**
>
> **Experimental verification.** To this end, we have further **conducted a new experiment** to compare the interactions extracted from two LLMs, i.e., LLaMA and Aquila-7B [cite 3].  As Figure 11 in Appendix N shows, two LLMs usually encode much more similar interactions than two not-so-large models. These experiments partially verify our hypothesis.
>
> [cite 2] Shen, W., et al. Can the Inference Logic of Large Language Models be Disentangled into Symbolic Concepts? arXiv preprint arXiv:2304.01083, 2023.
>
> [cite 3] BAAI. Aquila-7B, 2023. URL https://huggingface.co/BAAI/Aquila-7B.

---

> ### Author Response · Authors · 2023-11-18
>
> **Q3: Insights into the interpretability of extracted interactions.**
>
> > Q3: "Can the authors elaborate on whether the distinctive interactions extracted from more powerful models, such as BERT-large, result in enhanced interpretability?"
>
> **A3:** A good suggestion. We believe that the interaction is a better perspective to explain the representation similarity between different AI models than other metrics. It is because different AI models usually have different structures, and their features are not directly aligned to each other. It is a significant challenge to quantify the similarity of two DNNs. For example, let $f_A$ and $f_B$ denote the features of two networks A and B. Then, we learn an model $g$ to align $f_B$ to $f_A$, via $g=argmin \vert\vert f_A-g(f_B)\vert\vert^2$. Either $2\vert\vert f_A-g(f_B) \vert\vert / (\vert\vert f_A\vert\vert +\vert\vert g(f_B)\vert\vert)$, or $cos(f_A,g(f_B))$ still has certain technical flaws in evaluating the distance/similarity of two features. It is because we cannot theoretically guarantee that the numerical classification utilities of $f_A$, $g(f_B)$, and $f_A-g(f_B)$ are exactly proportional to their feature strength $\vert\vert f_A\vert\vert$, $\vert\vert g(f_B)\vert\vert$, and $\vert\vert f_A-g(f_B)\vert\vert$. Similarly, there is no theoretical guarantee for the faithfulness of $cos(f_A,g(f_B))$.
>
> In comparison, when we use interactions, [cite 4] have found that the network output can be faithfully decomposed into different interaction effects, no matter whether two DNNs have fully different architectures. Besides, because the interaction effect is directly responsible for network output, we do not face the problem that the feature strength is not the classification utility. **Therefore, we can use the similarity of interactions as the similarity of inference logics of two DNNs.**
>
> Furthermore, we have **conducted a new experiment** to compare the interactions extract from two LLMs, i.e., LLaMA and Aquila-7B [cite 3]. As Figure 11 in Appendix N shows, a pair of LLMs usually share more ratio of interactions than other pairs of language models. It means that most LLMs may usually converge to the same set of potentially true interactions, as long as the LLM has a huge number of parameters and is well trained on extensive data. We believe that our finding may help people interpret the common learning direction of different LLMs.
>
> [cite 3] BAAI. Aquila-7B, 2023. URL https://huggingface.co/BAAI/Aquila-7B.
>
> [cite 4] Li, M., et al. Does a neural network really encode symbolic concept? In ICML, 2023.

---

### Official Review · Reviewer_GVkW · 2023-10-31

**Soundness:** 2 fair
**Presentation:** 2 fair
**Contribution:** 3 good
**Rating:** 6
**Confidence:** 2

**Summary:**

The main challenge in explainable AI is to effectively summarize the knowledge encoded by a deep neural network (DNN) into a few symbolic primitive patterns without losing significant information. To tackle this, Ren et al. (2023) established a series of theorems demonstrating that a DNN's inference score can be explained as a small set of interactions between input variables. However, these interactions lack generalizability and thus cannot be considered as faithful primitive patterns encoded by the DNN. To address this, the authors propose  a method to extract interactions that are common to different DNNs trained for the same task.

**Strengths:**

- The paper is well-written and easy to follow.
- The experimental design is comprehensive, covering aspects of both NLP and vision, which provides a broad perspective.
- The paper focus on the interesting the generalization problem of interaction , which is a interesting problem.

**Weaknesses:**

- The paper does not sufficiently explain why generalizable interactions are important for AI interpretability. Although generalizability is critical in AI algorithms, especially when dealing with different tasks or data from different domains. However, the authors focus on explainable AI, and the generalizability is defined on different models of the same task, which does not align with the conventional understanding of DNN generalizability.
- The proofs in the paper rely on two papers currently under review ([1,2] See appendix C). This poses a problem as the validity and peer-review status of these references are uncertain. Moreover, it would be beneficial for the readers if detailed derivations of Equation 11 were provided to better understand and verify the proofs.
- While the paper provides a clear mathematical definition for generalizable interactions, it lacks a discussion on the existence of generalizable interactions. Moreover, the method proposed in the paper is derived heuristically using Occam's Razor principle instead of being grounded on mathematical theorems satisfying the definition. The paper does not satisfactorily address why this approach would yield generalizable interactions.
- The experimental section of the paper only compares the proposed method with a single baseline. Including more baselines could strengthen the paper's persuasiveness.

[1] Technical note: Defining and quantifying and-or interactions for faithful and concise explanation of dnns.

[2] Where we have arrived in proving the emergence of sparse symbolic concepts in ai models.

**Questions:**

See weekness.

---

> ### Author Response · Authors · 2023-11-18
>
> Thanks a lot for all the valuable comments and suggestions. We would like to answer all the questions and clarify your concerns. Besides, we have carefully revised the paper in blue according to your suggestions.
>
> **Please let us know as soon as possible, if you have any further questions or concerns.**
>
> **Q1: $\color{blue}{\textsf{Ask for the importance of generalizable interactions for AI interpretability.}}$**
>
> > Q1: "why generalizable interactions are important for AI interpretability" "the generalizability is defined on different models of the same task, which does not align with the conventional understanding of DNN generalizability."
>
> **A1:** Thank you. Unlike traditional studies on the generalization power of a DNN, **our research is led by a different motivation**, i.e., exploring the optimal way to explain inference logics of a DNN into symbolic interaction concepts. To this end, although the huge gap between symbolism and connectionism is well known, the counter-intuitive phenomenon of the sparsity of interactions in [cite 1] provides a new potential for the breakthrough of the seemingly impossible task. Specifically, previous studies [cite 1] [cite 2] have observed the sparsity of the interactions extracted in well-trained DNNs (in fact, this has been partially proven [cite 3] under three common conditions).
>
> **Therefore, our study of ensuring the interactions' generalization power may serve the last piece to prove/identify the symbolic/sparse interactions that faithfully explain a DNN.** If we simultaneously ensure that (1) each DNN just encodes a small number of salient interactions, comparing with all $2^n$ potential combinations in the input variables, (2) a small number of salient interactions can universally match the DNN outputs on all $2^n$ randomly masked sample $\\{\mathbf{x}_S|S\subseteq N\\}$, and (3) most interactions are generalizable over different DNNs trained for the same task, then we can consider that we have faithfully defined interaction concepts encoded by a DNN.
>
> The proof of a DNN’s encoding of symbolic interactions may help future analysis of generalization power of an AI model. First, **theoretically, we can redefine the generalization power of a DNN based on the learned interactions encoded by the DNN in future work.** In fact, the vanilla Harsanyi interaction has been used to explain the generalization power of a DNN [cite 4], in which low-order interactions have been found to have higher generalization power than high-order interactions. Other traditional studies mainly analyze generalization power in terms of the accuracy and the loss difference between training and testing samples. However, none of these metrics can reflect the intrinsic mechanism of the generalization power of a DNN. In comparison, the interaction concept provides us an alternative perspective to redefine the generalization of a DNN. Specifically, given multiple DNNs trained for the same task and an input sample, if an interaction can be extracted from all/most of these DNNs, then we consider it generalizable. In this way, we can identify a set of generalizable interactions and a set of non-generalizable interactions, which show much deeper insights into the generalization power of a DNN.
>
> [cite 1] Li, M., et al. Does a neural network really encode symbolic concept? In ICML, 2023.
>
> [cite 2] Ren, J., et al. Defining and quantifying the emergence of sparse concepts in dnns. In CVPR, 2023.
>
> [cite 3] Ren, Q., et al. Where we have arrived in proving the emergence of sparse symbolic concepts in ai models. arXiv preprint arXiv:2305.01939, 2023.
>
> [cite 4] Zhou, H., et al. Concept-Level Explanation for the Generalization of a DNN. arXiv preprint arXiv:2302.13091, 2023.
>
> **Q2: Asking for the proof of Theorem 3.**
>
> > Q2: "if detailed derivations of Equation 11 were provided" "The proofs in the paper rely on two papers currently under review ([1,2] See appendix C). "
>
> **A2:** Thank you very much for your careful review. Please see the revised derivations of the Equation (11) in Appendix D.
>
> First, the proposed loss function of learning generalizable interactions does **not** rely on the proof of two papers [cite 3] [cite 5]. We can still extract generalizable interactions without Theorem 3.
>
> Second, the sparsity of interactions has been widely observed in many studies [cite 1] [cite 2], so that we actually do **not** need the two under-review papers [cite 3] [cite 5] to support the design of our algorithm. To this end, we are sorry that we use the inaccurate statements that mislead the understanding of Theorem 3. We have revised the paper in Theorem 3 and Proposition 1.

---

> ### Author Response · Authors · 2023-11-18
>
> **A2:** Besides, the proposed universal-matching theorem on AND interactions has been proved in an CVPR paper [cite 2], as $\forall T\subseteq N, v(\mathbf{x}\_T)=\sum_{S\subseteq T}I\_{\text{\rm and}}(S|\mathbf{x})$. Then, we can prove the universal-matching theorem on both AND interactions and OR interactions (shown as the revised Theorem 3) in Appendix C. Then, according to the finding that well-trained DNNs tend to encode sparse interactions in [cite 1], we can also obtain the proposition $\forall T\subseteq N, v(\mathbf{x}\_T) \approx v(\mathbf{x}\_\emptyset) + \sum\nolimits_{\emptyset \neq S\subseteq T: S\in \Omega^{\text{\rm and}}}I\_{\text{\rm and}}(S|\mathbf{x}\_T) + \sum\nolimits_{S\cap T \neq \emptyset:S\in \Omega^{\text{\rm or}}}I_{\text{\rm or}}(S|\mathbf{x}_T)$. Given the findings in [cite 1], the proof of the sparsity of interactions in [cite 3] is no long so important to prove our conclusion.
>
> [cite 1] Li, M., et al. Does a neural network really encode symbolic concept? In ICML, 2023.
>
> [cite 2] Ren, J., et al. Defining and quantifying the emergence of sparse concepts in dnns. In CVPR, 2023.
>
> [cite 3] Ren, Q., et al. Where we have arrived in proving the emergence of sparse symbolic concepts in ai models. arXiv preprint arXiv:2305.01939, 2023.
>
> [cite 5] Li, M., et al.  Technical note: Defining and quantifying and-or interactions for faithful and concise explanation of dnns, 2023. arXiv preprint arXiv:2304.13312, 2023.
>
> **Q3:  "it lacks a discussion on the existence of generalizable interactions"**
>
> **A3:** Thank you. Theoretically, there is no guarantee that generalizable interactions necessarily exist. However, empirically, [cite 1] has observed that for many different tasks (e.g., image classification on the MNIST dataset and image classification on the CelebA-eyeglasses datasets), different DNNs trained for the same task all encode similar interactions. This is the motivation of this study, and we decide to go beyond empirical comparisons and formally design a loss function to learn generalizable interactions.
>
> In addition, we have **conducted more experiments** to show the generalization interactions between more DNNs (i.e., between the LLaMA and  Aquila-7B [cite 6]). Please see Appendix L for details.
>
> Although we cannot strictly prove that such generalization power of interactions is a generic property for all tasks, experiments show that most DNNs exhibit interactions with considerable generalization power. We believe this will provide new insights into the representation quality of a DNN.
>
> [cite 1] Li, M., et al. Does a neural network really encode symbolic concept? In ICML, 2023.
>
> [cite 6] BAAI. Aquila-7B, 2023. URL https://huggingface.co/BAAI/Aquila-7B.
>
> **Q4: Discussion on "the method proposed in the paper is derived heuristically using Occam's Razor principle"**
>
> > Q4:  "The paper does not satisfactorily address why this approach would yield generalizable interactions."
>
> **A4:** Thank you, but the generalization power of interactions is **not** guaranteed by the Occam’s Razor principle, although we do use this principle in Equation (6). Instead, we design a specific loss $\Vert\rm{rowmax}({I}\_{\text{\rm and}})\Vert_1 +\Vert\rm{rowmax}({I}\_{\text{\rm or}})\Vert_1$ in Equation (5), which exclusively penalizes non-generalizable interactions. Here, let us be given $m$ interactions on the same subset $S$, $I_{\rm and}^{(1)}(S|x), I_{\rm and}^{(2)}(S|x),...,I_{\rm and}^{(m)}(S|x)$, extracted from $m$ DNNs. We consider the interaction with the maximum absolute value $max_i \vert I_{\rm and}^{(i)}(S|x)\vert$ as the outlier effect that is exclusively contained by a single DNN. Thus, this loss only penalizes the outlier effect to boost the generalization power. For example, given $I^{(1)}(S|x)$ and $I^{(2)}(S|x)$ for the  same set of input variables $S\subseteq N$ extracted from two DNNs, if this interaction is non-generalizable, it usually means that the interactions extracted from two DNNs have significantly different effects on the model output. In this way, the loss in Equation (5) directly pushes the most salient interaction towards the other interactions, thereby making interactions in different DNNs similar to each other.
>
> We have revised the paper to clarify this in the paragraph under Equation (6).

---

> > ### Author Response · Authors · 2023-11-18
> >
> > **Q5: Asking to compare with more baselines.**
> >
> > > Q5: "Including more baselines could strengthen the paper’s persuasiveness."
> >
> > **A5:** Thank you. We have followed your suggestion to **conduct a new experiment** to compare our method with a new baseline. The new baseline is the original Harsanyi interaction used in [cite 2] (considering only the vanilla Harsanyi interaction in Equation (1)). In fact, most other interactions, such as Shapley interaction index [cite 4], do not satisfy the efficiency axiom (cannot ensure the network output as the sum of interactions), so these interactions cannot serve as competing methods of explaining compositional symbolic interaction primitives encoded by a DNN. We measured the sparsity of the interactions and the generalization power of the interactions extracted from different DNNs. The experimental results show that, compared to the new baseline, the interactions extracted by our method exhibit improved sparsity and generalization power. Please see Appendix J for details of the new experiments.
> >
> > [cite 2] Ren, J., et al. Defining and quantifying the emergence of sparse concepts in dnns. In CVPR, 2023.
> >
> > [cite 6] Grabisch, M. and Roubens, M.. An axiomatic approach to the concept of interaction among players in cooperative games. International Journal of Game Theory,1999.
> >
> >
> >
> > **Do our responses satisfy you? Please let us know as soon as possible, if you have any further questions or concerns.**

---

> > > ### Comment · Reviewer_GVkW · 2023-11-23
> > >
> > > Thnak you for answering the questions detailed, I appretiate the proof of Eq 11 and the new experiments. Most of my concerns are addressed. I will increase the score accorddingly.

---

> > > > ### Author Response · Authors · 2023-11-23
> > > >
> > > > Thank you very much for your appreciation.

---

### Official Review · Reviewer_f9us · 2023-11-01

**Soundness:** 2 fair
**Presentation:** 2 fair
**Contribution:** 2 fair
**Rating:** 5
**Confidence:** 4

**Summary:**

The paper begins by revisiting Harsayni's 1963 decomposition of Shapley values into "and" cohorts, and extends the idea to a claim that the output of a neural network can be decomposed into sparse contributions from "and" and "or" cohorts (Theorem 3).  Then it observes that such a composition could be ambiguous, and that it might not generalize to other models of the same data.  Then it proposes a framework for finding small and/or cohorts by optimizing for sparsity (eq 4), and further for maximizing sparsity while blending in another loss term that takes account sharing of variables in some way (eqs 5, 6).  Then it introduces a noising scheme (page 7) and proposes optimizing the search for cohorts in the presence of some level of noise.  Finally, the paper sets up experiments to search for such and/or cohorts within the behavior of a few large neural networks, examining extremely small subsets of the behavior of those network (reducing to 8-10 input variables), by manually selecting 8 patches within MNIST images of "3"s and 10 tokens for each set of inputs for SQuAD and SST-2 data sets, and examining the ability to find and/or cohorts that explain behavior of BERT, Llama, ResNet, etc.  It claims that with these methods it able to distinguish interactions that are shared between different models, and interactions that are in common.

**Strengths:**

The goal of trying to fully explain a neural network's behavior by decomposing it into just AND and OR interactions is admirable, and so is the ambition of attempting to apply such analysis to huge models such as Llama and BERT and VGG-16.

The focus on looking for commonalities between different models is interesting: it is an open and interesting question whether different models trained to solve the same task solve them in the same way or different ways, and how to understand the differences.

**Weaknesses:**

The paper does not refer to other work in examining shared features between networks, for example: https://arxiv.org/abs/2306.09346

The conclusions drawn from the paper are both noncommittal and unconvincing.  For example, the paper claims "The sparsity and universal-matching property of interactions provide lots of evidence to faithfully explain DNNs with interactions."  However, the methods are considered over only tiny subsets of just 8 or 10 features, (e.g., explained in appendix G, just 8 hand-chosen patches of images of MNIST "3" digits, or just 10 words out of the hundreds for each example in the NLP data sets).  As a result, what is being explained with these methods cannot be claimed to be faithful to "ResNet" or "BERT Large," but actually a very miniscule slice of the behavior of those huge networks.  The claim of faithfulness is not backed up empirically.

The methods are not clearly presented.  The main method comes down to Equation 6.  However most of the pages of the paper are spent on discussing Theorem 3, which is only tangentially related to the method and which does not clarify what the method actually does. For example, how continuous neural network features are discretized for the and-or analysis is not discussed at all.  How the search to optimize the discrete objective in Eq 6 is not discussed, and the level of sparsity, or the number of cohorts found in a typical solution is not revealed in concrete terms. If the goal is to clearly explain the method, it would be more helpful to move the tangential theorems to the appendix and spend more pages explaining equation 6 more precisely, with both concrete examples of how the interaction matrices are formatted, and concrete examples of how a search over cohorts would work in a specific small example.  Since runtime seems to be a major concern, it should be discussed. The size of the sets of features is small enough that it seems that a real-data example of how the method operates on a single example could be shown in detail.

The experiment results are very unclear, and it seems like it would be very difficult to reproduce the results given the information provided.  For example, in results in Fig 2 and 3 the y axis "Interaction Strength" is not explained other than that "it is in log space" or "it is the ratio of shared interactions".  What number is this this strength, and in what units? Beyond the lack of units, after the results are shown, they are not analyzed.  By analyzing the log-odds output of the ground-truth label, it looks like it is measuring faithfulness, but the plot captions claim a measurement of "interaction strength".  How interactions are being measured is not clear.  There is a very tiny 1/20 |I(s)|_max inside one of the plots that looks like a hint, but it is not explained.  Similarly, the units for Figure 4 are not explained.  All experiments are compared to a "Traditional" baseline which is not explained or summarized other than a reference to

Figure 5 shows some qualitative differences, but it is not clear what conclusions should be drawn from the differences.  For example, it shows that a disjunction "Ring just cold wet Seattle..." etc is promoted from the "Distinctive interactions" column using Traditional methods to the "Shared interactions" column in the Our method.  Is this a success or is this a failure?  If there are two different ways of identifying the interactions, does the new way provide more insight?  Or are they just different? Figure 5 select "some of the sailent" interactions - is it cherry-picked?  What does it look like when not?

**Questions:**

What is the runtime complexity of the method?
In the three data settings (or in the half dozen models) what are the actual numbers of cohorts of AND and OR variables that are found that achieve a specific high level of faithfulness?
How are features discretized for analysis?
What are the units of measure for interaction strength?
What can we conclude about the similarity between VGG and ResNet, or the similarity between BERT base and BERT large?

---

> ### Author Response · Authors · 2023-11-17
>
> Thank you very much for your comments, which help us much improve the paper in the rebuttal phase. It seems that most concerns (Q3, Q4, Q5, Q6, Q8) mainly focused on the clarity of the technical details. We have carefully revised the paper in blue according to your suggestions on the paper-writing issues. We have also carefully answered all other questions, and have conducted new experiments as you requested.
>
> **Please let us know as soon as possible, if you have any further questions or concerns.**
>
> **Q1: Discussion on the recommended paper in ICCV 2023.**
> >Q1: "refer to other work in examining shared features between networks"
>
> **A1:** Thank you. This paper is interesting. Unlike our objective of faithfully explaining DNNs into symbolic interactions, that paper identifies units (convolutional kernels) in different DNNs that share similar concepts. Beyond the analysis of feature maps of different units, our paper discovers that different DNNs trained for the same task tend to learn similar sets of interactions, as long as the different neural networks are sufficiently optimized, no matter whether these DNNs have different architectures and parameters. For example, as Figure 4 shows, two completely different LLMs surprisingly encode two similar sets of sparse interactions (phrases). Therefore, our attempt of extracting common interactions shared by different DNNs provides a new direction of pursuing the essential explanation of a DNN’s feature representations.
>
> Nevertheless, we have cited and discussed this paper in Appendix A in the revised paper.
>
> **Q2: $\color{blue}{\textsf{Asking about the evidence for the faithfulness of the sparsity and universal-matching.}}$**
> >Q2: Concerning the paper's claim that "The sparsity and universal-matching property of interactions …to faithfully explain DNNs with interactions"."the methods are considered over only tiny subsets of just 8 or 10 features …cannot be claimed to be faithful to 'ResNet' or 'BERT Large' ", "The claim of faithfulness is not backed up empirically."
>
> **A2:** It is a very good question. Let us clarify both in the theoretical proofs and in experiments for the faithfulness of the sparsity and universal-matching.
>
> Theoretically, the faithfulness means that given an input with $n$ variables, we must prove that (1) the DNN only encodes a small number of salient interactions $\Omega$, comparing with all $2^n$ potential combinations of the input variables, i.e., $\vert\Omega\vert \ll 2^n$, and that (2) the network outputs on all $2^n$ randomly masked samples $\\{\mathbf{x}_S|S\subseteq N\\}$ can be well matched by a few interactions in $\Omega=\\{S\subseteq N: \vert I(S|x) \vert > \tau\\}$, $\Omega$ is defined in the eighth paragraph in Section 2.1 in the revised paper. These two terms have been proved in Theorem 1, Theorem 3 and Proposition 1 in the revised paper.
>
> Then, in practice, considering the $2^n$ computational complexity, we have followed settings in [cite 1] to extract interactions between a set of randomly selected input variables $N=\\{1,2,…,t\\}, (t<n)$ (other unselected $(n-t)$ input variables will remain unmasked, leaving the original state unchanged). In this case, faithfulness does not mean that our interactions explain all the inference logic encoded between all $n$ input variables for a given sample $x$ in the pre-trained DNN. Instead, it only means that the extracted interactions can also accurately match the inference logic encoded between the selected $t$ input variables for a given sample $x$ in the DNN. We have clarified this in Appendix G in the revised paper.
>
> **Experimental verification.** In addition, we have further conducted **a new experiment** to verify the faithfulness when we explain interactions between all input variables. To this end, for the sentiment classification task on the SST-2 dataset in ${\rm BERT_{BASE}}$, we selected sentences containing 15 tokens to verify the faithfulness of the sparsity in Proposition 1 and the universal-matching property in Theorem 3. All tokens are selected as input variables.  We focused on the matching errors $\epsilon_T=v^{\rm approx}(x_T) - v(x_T), v^{\rm approx}(x_T):=v(x_{\emptyset})+ \sum_{\emptyset\ne S\subseteq T : S\in \Omega^{\rm and}}I_{\text{\rm and}}(S|x_T) + \sum_{ S\cap T \neq \emptyset: S\in \Omega^{\text{\rm or}}}I_{\rm or}(S|x_T)$ for all masked samples  $x_T, \forall T\subseteq N$. Specifically, we observed whether the real network output on the masked sample $v(x_T ), \forall T\subseteq N$ can be well approximated by interactions. We have verified that the extracted salient interactions in $\Omega$ faithfully explain the model output, i.e., $\forall T\subseteq N, v(x_T) \approx v(x_{\emptyset})+ \sum_{\emptyset\ne S\subseteq T : S\in \Omega^{\rm and}}I_{\text{\rm and}}(S|x_T) + \sum_{ S\cap T \neq \emptyset: S\in \Omega^{\text{\rm or}}}I_{\rm or}(S|x_T)$. Please see Appendix G  in the revised paper for more details.

---

> ### Author Response · Authors · 2023-11-18
>
> **A2:**
> [cite 1] Li, M., et al. Does a neural network really encode symbolic concept? In ICML, 2023.
>
> **Q3: $\color{blue}{\textsf{About paper writing.}}$**
>
> > Q3: "how continuous neural network features are discretized for the and-or analysis"
>
> **A3:** Thank you. We have followed your suggestion to clarify how to define discretized interactions for continuous input variables in blue in the footnote [3]. We did not discretize the continuous values of each input variable. Instead, we simply followed [cite 1] to mask each input variable $i\in N\setminus S$ by defining and using baseline values (see Appendix C for the definition of the baseline values in the revised paper). For example, for the image dataset, we used zero as the baseline value to represent the masked state of the input variable in $i\in N\setminus S$ (see Appendix M.2 for experimental details in the revised paper). In this way, for each input sample with $n$ input variables, there are a total of $2^n$ different masked states $\\{x_S\\}$, as described in the tenth paragraph of Section 2.1. Therefore, the masked and unmasked states are two discretized states for each input variable, so that we can define the discretized interactions  $I_{\rm and}$ and $I_{\rm or}$ based on the masked samples in Equations (1) and (2).
>
> [cite 1] Li, M., et al. Does a neural network really encode symbolic concept? In ICML, 2023.
>
> **Q4: $\color{blue}{\textsf{About paper writing. Ask for detailed explanation for Equation 6.}}$**
>
> > Q4: "How the search to optimize the discrete objective in Eq 6 is not discussed, and the level of sparsity" "the number of cohorts found in a typical solution is not revealed in concrete terms" "with both concrete examples of how the interaction matrices are formatted … in a specific small example" "a real-data example of how the method operates on a single example could be shown in detail"
>
> **A4:** Thank you. Please see answers to Q3 for the concerns on the discretization problem. Then, let us answer other concerns as follows.
>
> $\bullet$ The optimization of the loss function in Equation (6) is simply based on the gradient descent method. This is because the parameter $\gamma_{T}\in R$ to be optimized is continuous, rather than discrete. As mentioned in the first paragraph of Section 2.3, we learned the optimal values of $\gamma_{T}, T\subseteq N$ to decompose the model output $v(x_T)$ into two terms $v_{\rm and}(x_T)$ and $v_{\rm or}(x_T)$. The loss was designed to boost the sparsity of interactions, which ensured the sparsity of interactions. That is, such optimal values of $\gamma_{T}$ are learned to make most interactions almost zero effect, $I_{\rm and}(T|x) \approx 0$ and $I_{\rm or}(T|x) \approx 0$.
>
> $\bullet$ **The number of interactions been optimized.** During the optimization process, the loss in Equation (6) is applied to all $2^n$ interactions. After  the learning of parameters $\\{ \gamma_T | T\subseteq N\\}$, we compute the $2^n$ interactions based on Equations (1) and (2). Most of these interactions have almost zero effect. We select interactions which are greater than the threshold $\tau$ as salient interactions. The final number of salient interactions is determined by the threshold $\tau$. Please see the eighth paragraph of Section 2.1 for details.
>
> Besides, we have also **conducted experiments** to show the number of salient interactions extracted by optimizing the loss in Equation (6). For example, Figure 2 shows that Equation (6) extracts 2915 salient interactions  ($\approx$ 0.71%)  from the $(2\cdot2^{10})\times200$ AND-OR interactions extracted from a total of 200 samples in the ${\rm BERT_{BASE}}$ model, when we set the threshold $\tau=1/20\cdot\max_S\vert I(S|x)\vert$.
>
> $\bullet$ The interaction matrices in Equation (6) are formulated in the first sentence below Equation (5).
>
> Let us use a concrete example to illustrate how the interaction matrices are formatted in a toy example. Let us consider two pre-trained DNNs $v^{(1)}$, $v^{(2)}$ and an input sample $x$ with $N=\\{1,2\\}$ variables, and take the AND interaction matrix ${I}\_{\rm and}$ in Equation (6) as an example (the OR interaction matrix can be obtained similarly). First, we get all $2^2$ AND interactions extracted from the $i$-th model $\mathbf{I}^{(i)}\_{\rm and}=[ I\_{\rm and}(T_1|x), I\_{\rm and}(T_2|x), I\_{\rm and}(T_3|x), I\_{\rm and}(T_4|x)]^\intercal \in \mathbb{R}^{2^2}$, according to the description of Equation (4). Here, each interaction value $I_{\rm and}(T_k|x) \in \mathbb{R}, T_k \subseteq N$ denotes the interaction value of each masked sample $x_{T_k}$ and is computed by Equation (1). Second, the AND interaction matrix ${I}\_{\rm and}= \left[ \mathbf{I}^{(1)}\_{\rm and}, \mathbf{I}^{(2)}_{\rm and} \right] \in \mathbb{R}^{2^2\times 2}$ denotes the interaction values corresponding to the $2^2$ masked samples for each of the two models.

---

> > ### Author Response · Authors · 2023-11-18
> >
> > **A4:** Let us use a concrete example to illustrate how a search over cohorts would work.  The loss in Equation (6) in the above example can be represented as the function of $\\{\gamma_T\\}$ as follows. $Loss = \min\nolimits_{\ \\{\gamma\_{T\_k}^{(1)}, \gamma\_{T\_k}^{(2)}\\}_{k}\}\left( \Vert\rm{rowmax}({I}\_{\text{\rm and}}) \Vert_1 + \Vert\rm{rowmax}({I}\_{\text{\rm or}})\Vert_1\right)+\alpha\left( \Vert{I}\_{\text{\rm and}}\Vert_1 +\Vert{I}\_{\text{\rm or}}\Vert_1 \right)$
> >
> > $=\min\nolimits_{\\{\gamma\_{T_k}^{(1)},\gamma\_{T_k}^{(2)}\\}_{k}}\sum\_{T_k\subseteq N} \vert \max(\sum\nolimits\_{T \subseteq S}(-1)^{|S|-|T|} [0.5\cdot v^{(1)}(\mathbf{x}\_{T_k})+\gamma\_{T_k}^{(1)}], \sum\nolimits\_{T \subseteq S}(-1)^{|S|-|T|}[0.5\cdot v^{(2)}(\mathbf{x}\_{T_k})+\gamma\_{T_k}^{(2)}])\vert$
> >
> > $+\sum\_{T_k\subseteq N} \vert \max(-\sum\nolimits\_{T \subseteq S}(-1)^{|S|-|T|} [0.5\cdot v^{(1)}(\mathbf{x}\_{N\setminus T_k})-\gamma\_{T_k}^{(1)}], -\sum\nolimits\_{T \subseteq S}(-1)^{|S|-|T|}[0.5\cdot v^{(2)}(\mathbf{x}\_{N\setminus T_k})-\gamma\_{T_k}^{(2)}])\vert$
> >
> > $+\alpha \cdot \sum_{T_k\subseteq N}\sum_{i=1}^{2} \vert\sum\nolimits\_{T \subseteq S}(-1)^{|S|-|T|}[0.5\cdot v^{(i)}(\mathbf{x}\_{T_k})+\gamma_{T_k}^{(i)}]\vert$
> >
> > $+ \alpha \cdot \sum_{T_k\subseteq N}\sum_{i=1}^{2}\vert -\sum\nolimits\_{T \subseteq S}(-1)^{|S|-|T|}[0.5\cdot v^{(i)}(\mathbf{x}\_{N\setminus T_k})-\gamma_{T_k}^{(i)}])\vert $
> >
> > Therefore, we only need to optimize  $\\{\gamma_{T_k}^{(1)}, \gamma_{T_k}^{(2)}\\}_k$ via gradient descent in Equation (6).
> >
> > **Q5: $\color{blue}{\textsf {About paper writing. Ask for more explanations for experiment results.}}$**
> >
> > > Q5: "the y axis 'Interaction Strength' is not explained" "What number is this this strength, and in what units?" "after the results are shown, they are not analyzed" "it looks like it is measuring faithfulness … How interactions are being measured is not clear" "1/20 |I(s)|_max inside one of the plots that looks like a hint, but it is not explained" "the units for Figure 4 are not explained"
> >
> > **A5:** Thank you for your careful comments. We would like to clarify all these details.
> >
> > $\bullet$ We have revised Figure 2 to improve the clarity according to your comments. In Figure 2, the 'Interaction Strength' represents the absolute value of interaction, i.e., $ \vert I(S|x)\vert, S\subseteq N$. The interaction strength measures the absolute value of the interaction effect on the network output according to Equations (1) and (2), so the unit of interaction strength is the same as the network output $v(x)= \log\frac{p(y=y^{\text{truth}}|\mathbf{x})}{1-p(y=y^{\text{truth}}|\mathbf{x})}$ (see Task 1 in Section 3). Similarly, the y-axis of Figure 4 shows four metrics. $\textit{All}^{+, (i)}(o)=\sum_{\text{\rm op}\in\\{\text{\rm and},\text{\rm or}\\}}\sum_{S\in \Omega^{\text{\rm op}, (i)} ,|S|=o}\max(0,I_\text{\rm op}^{(i)}(S|\mathbf{x}))$  and
> > $\textit{All}^{-, (i)}(o)=\sum_{\text{\rm op}\in\\{\text{\rm and},\text{\rm or}\\}}\sum_{S\in \Omega^{\text{\rm op}, (i)} ,|S|=o}\min(0,I_\text{\rm op}^{(i)}(S|\mathbf{x}))$ represent the overall strength of salient positive interactions and that of salient negative interactions of the $i$-th DNN, respectively. $\textit{Shared}^{+, (i)}(o)=\sum_{\text{\rm op}\in\\{\text{\rm and},\text{\rm or}\\}}\sum_{S\in \Omega_{\text{\rm shared}}^{\text{\rm op}, (i)} ,|S|=o}\max(0,I_\text{\rm op}^{(i)}(S|\mathbf{x}))$ and $\textit{Shared}^{-, (i)}(o)=\sum_{\text{\rm op}\in\\{\text{\rm and},\text{\rm or}\\}}\sum_{S\in \Omega_{\text{\rm shared}}^{\text{\rm op}, (i)} ,|S|=o}\min(0,I_\text{\rm op}^{(i)}(S|\mathbf{x}))$ represent the overall strength of all positive interactions and that of all negative interactions of the $i$-th DNN,
> > which were shared by other DNNs. Please see the seventh paragraph in Section 3 for the definition of these metrics.
> >
> > In the description of Figure 2 (see 'Sparsity of the extracted primitives' in Section 3), we have mentioned that we set a threshold $\tau^{(i)}$ to collect a set of salient interactions (see 'Definition of interaction primitives'  in Section 2.1) from different DNNs, i.e., $\tau^{(i)}=1/20\cdot \max_S\vert I(S|x)\vert$.
> >
> > Figure 2 does not directly measure faithfulnes. Instead, it shows that the interactions extracted by our method are also very sparse compared to the traditional method. Only  very few interactions are larger than the threshold $\tau^{(i)}=1/20\cdot \max_S\vert I(S|x)\vert$, which are considered salient interactions (see 'Definition of interaction primitives'  in Section 2.1). Most of the interactions are far smaller than the threshold $\tau^{(i)}$, and therefore have negligible effect on the model output.

---

> > > ### Author Response · Authors · 2023-11-18
> > >
> > > **A5:** $\bullet$  We have revised Figure 3 to improve the clarity according to your comments. In Figure 3, the 'generalization power (ratio of shared interactions)' represents the ratio of shared salient interactions in different DNNs, i.e., $s_{\text{\rm{and}}}^{(i)}=|\Omega_{\text{\rm{shared}}}^{\text{\rm{and}}}|/|\Omega^{\text{\rm{and}},(i)}|$ and $s_{\text{\rm{or}}}^{(i)} =|\Omega_{\text{\rm{shared}}}^{\text{\rm{or}}}|/|\Omega^{\text{\rm{or}},(i)}|$ in Definition 1  in Section 2.2. Figure 3 shows the interactions extracted from our method exhibit higher generalization power (higher ratio of shared interactions in different DNNs) than interactions extracted from the traditional method.
> > >
> > > Please see the revised Figures 2 and 3 for details.
> > >
> > > **Q6: $\color{blue}{\textsf{About paper writing.}}$**
> > > >Q6: Details for "a 'Traditional' baseline which is not explained or summarized"
> > >
> > > **A6**: Thanks. We have added more discussions  to introduce traditional method in the fifth paragraph of Section 3. The traditional method is proposed to learn the sparest interactions in Equation (4), without considering the generalization between interactions extracted from multiple DNNs.
> > >
> > > **Q7: More discussion on Figure 5.**
> > > >Q7: "some qualitative differences … what conclusions should be drawn from the differences" "Is this a success or is this a failure? … does the new way provide more insight?" "some of the salient" interactions - is it cherry-picked?"
> > >
> > > **A7**: Thank you. We would like to follow your suggestion to emphasize the conclusion from Figure 5. Our method extracted much more shared interactions than the traditional interaction-extraction method, which shows that our method could obtain more stable explanation of the inference logics of a DNN. It is because the interactions shared by different DNNs were usually considered more faithful. We have added more discussions on this in the last paragraph of Section 3.
> > >
> > > Besides, "some of the salient" interactions were randomly selected, because the number of salient interactions was too large to be all shown in Figure 5. To this end, we present more salient interactions in the Appendix M.
> > >
> > > **Q8: $\color{blue}{\textsf{Ask about the runtime complexity.}}$**
> > > > Q8: "What is the runtime complexity of the method?" "Since runtime seems to be a major concern, it should be discussed."
> > >
> > > **A8**: Thank you very much for your suggestion. Theoretically, given an input sample with $n$ input variables, the time complexity is $2^n$, and we need to generate $2^n$ masked samples for model inference. Fortunately, the variable number is not too large by using the annotation of input variables provided in [cite1] [cite2]. In real applications,  the average running time for  a sentence in the SST-2 dataset on the Bert-large model is 45.14 seconds. In addition, real applications usually just require to annotate a small set of important input variables and compute interactions between important input variables, which significantly alleviates the computational burden.
> > >
> > > [cite 1] Li, M., et al. Does a neural network really encode symbolic concept? In ICML, 2023.
> > >
> > > [cite 2] Shen, W., et al. Can the Inference Logic of Large Language Models be Disentangled into Symbolic Concepts? arXiv preprint arXiv:2304.01083, 2023.
> > >
> > > **Q9: "What can we conclude about the similarity between VGG and ResNet, or the similarity between BERT base and BERT large?"**
> > >
> > > **A9**: Thanks. We have clarified this in the seventh paragraph of Section 3. The high similarity of interactions between VGG-16 and ResNet-20 shows that although two DNNs for the same tasks may have fully different architectures, there may exist a set of ultimate interactions for a task. Different well-optimized DNNs may be all optimized towards such interactions.
> > >
> > > **Do our responses satisfy you? Please let us know as soon as possible, if you have any further questions or concerns.**

---

> > ### Comment · Reviewer_f9us · 2023-11-22
> >
> > Several things remain unclear in the paper.
> >
> > * Exactly what is the procedure that is being done in the paper?  Can you give a concrete example of the experimental procedure from beginning to end for one of your small experimental settings?
> > * The existing text does not make it clear.  For example, despite the rebuttal assertion, Appendix C does not define baseline values.  It also does not clearly define the masking procedure, for example, there is no variable for the mask nor is there a formula describing the relationship between the mask and x and b.  An end-to-end example on a simplified case (e.g., mnist) might be helpful to clarify.
> > * Similarly, appendix M.2 does not seem to exist in the revision.  The additional examples in Figure 10 are just the same as the previously given ones and do not add further clarity.
> >
> > The paper's extraordinary claim, i.e., that DNNs can be analyzed as simple binary predicate and/or logic, requires extraordinary care, clarity and evidence, but the current paper does not meet that standard.
> >
> > The paper would be improved it were restructured, with a clear formulation section defining all the notation and the experimental procedure from beginning to end, with an example.  However, that would be a more significant rewrite than has currently been done.
> >
> > Score remains a 5.

---

> > > ### Author Response · Authors · 2023-11-22
> > >
> > > Thank you very much for your careful comments. We will respond to your concerns and questions within 24 hours.

---

> > > ### Author Response · Authors · 2023-11-23
> > >
> > > **Q1: Using a concrete example to illustrate the procedure.**
> > >
> > > > "Exactly what is the procedure that is being done in the paper? Can you give a concrete example of the experimental procedure from beginning to end for one of your small experimental settings?", "The existing text does not make it clear. For example, despite the rebuttal assertion, Appendix C does not define baseline values. It also does not clearly define the masking procedure, for example, there is no variable for the mask nor is there a formula describing the relationship between the mask and x and b. An end-to-end example on a simplified case (e.g., mnist) might be helpful to clarify."
> > >
> > > **A1:** Thank you. Let us use a concrete input sentence with six tokens to illustrate the experimental procedure from beginning to end.
> > >
> > > **Step 1:** Given the sentence "A stitch in time saves nine" with six tokens, the six input variables are $x_1=$ embedding of token "A",  $x_2=$ embedding of token "stitch", $x_3=$ embedding of token "in", $x_4=$ embedding of token "time", $x_5=$ embedding of token "saves", and $x_6=$ embedding of token "nine".  Although the dimensions of the embedding for the  ${\rm BERT_{BASE}}$ and  ${\rm BERT_{LARGE}}$ models are different, we can still compute the interactions between the embeddings corresponding to the same token. Specifically, the embedding $x_i$ of a token in the ${\rm BERT_{BASE}}$ model is $x_i\in\mathbb{R}^{768}$, and the embedding $x_i$ of a token in the ${\rm BERT_{LARGE}}$ model is $x_i\in\mathbb{R}^{1024}$.
> > >
> > > **Step 2:** The baseline value for each input variable is $b_1=b_2=b_3=b_4=b_5=b_6=$ special embedding of a masked token, where the special embedding of the masked token is encoded by the ${\rm BERT_{BASE}}$ and ${\rm BERT_{LARGE}}$ model, respectively.  This special embedding can use the embedding of a special token, e.g., the embedding of the [CLS] token. Specifically, the baseline $b_i$ in the ${\rm BERT_{BASE}}$ model is $b_i\in\mathbb{R}^{768}$, and the baseline $b_i$ in the ${\rm BERT_{LARGE}}$ model is $b_i\in\mathbb{R}^{1024}$.
> > >
> > > **Step 3:** In this case, there are a total of $2^6$ masked samples $\mathbf{x}\_{T_0},\mathbf{x}\_{T_1}, \mathbf{x}\_{T_2}, \mathbf{x}\_{T_3},\cdots, \mathbf{x}\_{T_{62}}, \mathbf{x}\_{T _{63}}$ used for the model inference. Specifically, the first masked sample is $\mathbf{x}\_{T_0}=\\{b_1, b_2, b_3, b_4, b_5, b_6\\}$,  where each of its input variables (embedding) is replaced with the corresponding baseline value (embedding of a special token), i.e., $x_i = b _i, i\in\\{1,2,\cdots,6\\}$. The second masked sample is $\mathbf{x}\_{T_1}=\\{b_1, b_2, b_3, b_4, b_5, x_6\\}$, where its input variable $x_6$ is kept as the original embedding, and the other five input variables are replaced with the corresponding baseline values, i.e., $x_1 = b_1, x_2 = b_2, x_3 = b_3, x_4 = b_4, x_5= b_5$. The third masked sample is $\mathbf{x}\_{T_2}=\\{b_1, b_2, b_3, b_4, x_5, b_6\\}$, where its input variable $x_5$ is  kept as the original embedding, and the other five input variables are replaced with the corresponding baseline values, i.e., $x_1 = b_1, x_2 = b_2, x_3 = b_3, x_4 = b_4, x_6= b_6$. The fourth masked sample is $\mathbf{x}\_{T_3}=\\{b_1, b_2, b_3, b_4, x_5, x_6\\}$, where its input variables $x_5$ and $x_6$ are kept as the original embedding, respectively, and the other four input variables are replaced with the corresponding baseline values.
> > >
> > > Similarly, the 63rd masked sample is $\mathbf{x}\_{T_{62}}=\\{x_1, x_2, x_3, x_4, x_5, b_6\\}$, where its input variables  $x_1$, $x_2$, $x_3$, $x_4$, $x_5$ are kept as the original embedding, respectively, and $x_6 = b_6$. The  64th masked sample is $\mathbf{x}\_{T_{63}}=\\{x_1, x_2, x_3, x_4, x_5, x_6\\}$, where all input variables are kept unchanged from the original embedding.
> > >
> > > **Step 4:** For each masked sample $\mathbf{x}\_{T_j}, j\in\\{0,1,\cdots,63\\}$,  we computed the log-odds output of the ground-truth label $v(\mathbf{x}\_{T_j})=\log\frac{p(y=y^{\rm truth}|\mathbf{x}\_{T_j})}{1-p(y=y^{\rm truth}|\mathbf{x}\_{T_j})}\in \mathbb{R}$ as the model output. In this way, feeding all masked samples  $\mathbf{x}\_{T_j}, j\in\\{0,1,\cdots,63\\}$ into the ${\rm BERT_{BASE}}$ model produced a total of $2^6$ model outputs $v^{\rm BASE}(\mathbf{x}\_{T_0})$, $v^{\rm BASE}(\mathbf{x}\_{T_1})$, $\cdots$,$v^{\rm BASE}(\mathbf{x}\_{T_{63}})$. Feeding all masked samples  $\mathbf{x}\_{T_j}, j\in\\{0,1,\cdots,63\\}$ into the ${\rm BERT_{LARGE}}$ model produced a total of $2^6$ model outputs $v^{\rm LARGE}(\mathbf{x}\_{T_0})$, $v^{\rm LARGE}(\mathbf{x}\_{T_1})$, $\cdots$,$v^{\rm LARGE}(\mathbf{x}\_{T_{63}})$.

---

> > > > ### Author Response · Authors · 2023-11-23
> > > >
> > > > **A1:**  **Step 5:**  Computed the AND outputs $v^{\rm BASE}\_{\rm and}(\mathbf{x}\_{T_j})=0.5v^{\rm BASE}(\mathbf{x}\_{T_j})+\gamma^{\rm BASE}\_{T_j}, j\in\\{0,1,\cdots,63\\}$ and the OR outputs $v^{\rm BASE}\_{\rm or}(\mathbf{x}\_{T_j})=0.5v^{\rm BASE}(\mathbf{x}\_{T_j})-\gamma^{\rm BASE}\_{T_j}, j\in\\{0,1,\cdots,63\\}$ for the ${\rm BERT_{BASE}}$ model, respectively. Computed the AND outputs $v^{\rm LARGE}\_{\rm and}(\mathbf{x}\_{T_j})=0.5v^{\rm LARGE}(\mathbf{x}\_{T_j})+\gamma^{\rm LARGE}\_{T_j}, j\in\\{0,1,\cdots,63\\}$ and the OR outputs $v^{\rm LARGE}\_{\rm or}(\mathbf{x}\_{T_j})=0.5v^{\rm LARGE}(\mathbf{x}\_{T_j})-\gamma^{\rm LARGE}\_{T_j}, j\in\\{0,1,\cdots,63\\}$ for the ${\rm BERT\_{LARGE}}$ model, respectively.
> > > >
> > > > **Step 6:** Computed the AND interactions $ I^{\rm BASE}\_{\text{and}}(T_j|\mathbf{x}) =  \sum\nolimits_{T \subseteq T_j}(-1)^{|T_j|-|T|}v^{\rm BASE}\_{\rm and}(\mathbf{x}\_T), j\in\\{0,1,\cdots,63\\}$ and the OR interactions $I^{\rm BASE}\_{\text{or}}(T_j|\mathbf{x}) =  -\sum\nolimits_{T \subseteq T_j}(-1)^{|T_j|-|T|}v^{\rm BASE}\_{\rm or}(\mathbf{x}\_{N\setminus T}), j\in\\{0,1,\\cdots,63\\}$ for the ${\rm BERT_{BASE}}$ model, respectively. Computed the AND interactions $ I^{\rm LARGE}\_{\text{and}}(T_j|\mathbf{x}) =  \sum\nolimits\_{T \subseteq T_j}(-1)^{|T_j|-|T|}v^{\rm LARGE}\_{\rm and}(\mathbf{x}\_T), j\in\\{0,1,\cdots,63\\}$ and the OR interactions $I^{\rm LARGE}\_{\text{or}}(T_j|\mathbf{x}) =  -\sum\nolimits\_{T \subseteq T_j}(-1)^{|T_j|-|T|}v^{\rm LARGE}\_{\rm or}(\mathbf{x}\_{N\setminus T}), j\in\\{0,1,\cdots,63\\}$ for the ${\rm BERT_{LARGE}}$ model, respectively.
> > > >
> > > > **Step 7:** Learned the parameters $\gamma^{\rm BASE}\_{T_j}$ and $\gamma^{\rm LARGE}\_{T_j}, j\in\\{0,1,\cdots,63\\}$ using the loss in Equation (6). Went back to Step 5 and repeated the iterations until the loss converges.
> > > >
> > > > **Step 8:** Computed the final AND interactions $ I^{\rm BASE}\_{\text{and}}(T_j|\mathbf{x})$ and the final OR interactions $I^{\rm BASE}\_{\text{or}}(T_j|\mathbf{x}), j\in\\{0,1,\cdots,63\\}$ for the ${\rm BERT\_{BASE}}$ model using the learnable parameters  $\gamma^{\rm BASE}\_{T_j},j\in\\{0,1,\cdots,63\\}$. The final salient interactions of the ${\rm BERT_{BASE}}$ model are obtained from all $2^6$ interactions, where the final salient interactions are those with interaction values greater than the threshold $\tau^{\rm BASE}$, $\Omega^{\rm BASE}=\\{T_j\subseteq N: \vert I^{\rm BASE}_{\text{and/or}}(T_j|\mathbf{x}) \vert > \tau^{\rm BASE}\\}$.
> > > >
> > > > Computed the final AND interactions $ I^{\rm LARGE}\_{\text{and}}(T_j|\mathbf{x})$ and the final OR interactions $I^{\rm LARGE}\_{\text{or}}(T_j|\mathbf{x}), j\in\\{0,1,\cdots,63\\}$  for the ${\rm BERT_{LARGE}}$ model using the learnable parameters  $\gamma^{\rm LARGE}\_{T_j},j\in\\{0,1,\cdots,63\\}$. The final salient interactions of the ${\rm BERT_{LARGE}}$ model are obtained from all $2^6$ interactions, where the final salient interactions are those with interaction values greater than the threshold $\tau^{\rm LARGE}$,  $\Omega^{\rm LARGE}=\\{T_j\subseteq N: \vert I^{\rm LARGE}_{\text{and/or}}(T_j|\mathbf{x}) \vert > \tau^{\rm LARGE}\\}$.
> > > >
> > > > We have added this concrete example in Appendix O.
> > > >
> > > > **Q2: "Similarly, appendix M.2 does not seem to exist in the revision. The additional examples in Figure 10 are just the same as the previously given ones and do not add further clarity."**
> > > >
> > > > **A2:** We are sorry that we used the wrong location 'Appendix M.2'. Instead, the experimental details for baseline values are introduced in the last paragraph of Appendix Q.2 in the revision paper.
> > > >
> > > > We provided additional clarification to Figure 10. Specifically, we randomly selected 10 tokens in the given sentence,  labeling these 10 tokens in red and the other unselected tokens in black. Here, $n=10$ input variables denote the embeddings corresponding to these 10 tokens. Since each input variable has two states, masked and unmasked,  a total of $2^{10}$ masked samples are generated. Then, according to Equations (1) and (2), a total of $2 \cdot 2^{10}$ AND interactions and OR interactions are finally obtained.

---

> > > > > ### Author Response · Authors · 2023-11-23
> > > > >
> > > > > **A2:** Then, we extracted the most salient $k=50$ interactions from a total of $2 \cdot 2^{10}$ AND interactions and OR interactions as the set of AND-OR interaction primitives in each DNN, respectively. Therefore, in our proposed method, 25 shared interactions were extracted from both models, and 25 distinctive interactions were extracted from the  ${\rm BERT_{BASE}}$ and ${\rm BERT_{LARGE}}$ models, respectively .  In contrast, in the traditional method, 16 shared interactions were extracted from  both models, and 34 distinctive interactions were extracted from  the  ${\rm BERT_{BASE}}$ and ${\rm BERT_{LARGE}}$ models, respectively. In addition, Figure 10 shows the strength of the interaction value for each salient interaction, i.e., $|I(S|\mathbf{x})|$. Figure 10 illustrates that, our method extracted more shared interactions compared to the traditional interaction-extraction method.
> > > > >
> > > > >
> > > > >
> > > > > **Q3: "The paper's extraordinary claim, i.e., that DNNs can be analyzed as simple binary predicate and/or logic, requires extraordinary care, clarity and evidence, but the current paper does not meet that standard."**
> > > > >
> > > > > **A3:** Thank you. Let us illustrate the evidence for faithfully explaining DNNs with AND interactions and OR interactions, respectively.
> > > > >
> > > > > First, the proposed universal-matching theorem on AND interactions has been proved in an CVPR paper [cite 1], which used the AND interactions to faithfully explain the outputs of DNNs, as $\forall T\subseteq N, v(\mathbf{x}\_T)=\sum_{S\subseteq T}I_{\text{\rm and}}(S|\mathbf{x})$. Specifically, [cite 1] have used the Harsanyi dividend $I_{\text{\rm and}}(S|\mathbf{x})$ [cite 2] to state the  universal matching theorem of AND interactions, in which the Harsanyi dividend $I_{\text{\rm and}}(S|\mathbf{x})$ can be composed into the Shapley value $\phi(i|\mathbf{x})$, i.e., $\phi(i|\mathbf{x}) = \sum_{S\subseteq N:S\ni i}\frac{1}{|S|}I_{\text{\rm and}}(S|\mathbf{x})$, see Theorem 2 for the details.
> > > > >
> > > > > Second, we have proved the universal-matching theorem on both AND interactions and OR interactions (shown as the revised Theorem 3) in Appendix C, as $\forall T\subseteq N, v(\mathbf{x}\_T) =   v\_{ \text{\rm and}}(\mathbf{x}\_T) + v\_{\text{\rm or}}(\mathbf{x}\_T) =  \sum\nolimits_{ S\subseteq T }I\_{\text{\rm and}}(S|\mathbf{x}\_T) + \sum\nolimits_{S\in \\{S: S\cap T \neq \emptyset\\}\cup\\{\emptyset\\}}I\_\text{\rm or}(S|\mathbf{x}\_T).$
> > > > >
> > > > > Third, according to the finding that well-trained DNNs tend to encode sparse interactions in [cite 1] [cite 3], we can also obtain the proposition $\forall T\subseteq N, v(\mathbf{x}\_T) \approx v(\mathbf{x}\_\emptyset) + \sum\nolimits_{\emptyset \neq S\subseteq T: S\in \Omega^{\text{\rm and}}}I\_{\text{\rm and}}(S|\mathbf{x}\_T) + \sum\nolimits_{S\cap T \neq \emptyset:S\in \Omega^{\text{\rm or}}}I\_{\text{\rm or}}(S|\mathbf{x}\_T)$ in the Proposition 1.
> > > > >
> > > > > Besides, [cite 3] also empirically demonstrated that well-trained DNNs usually encode transferable and discriminative interactions on extensive tasks.
> > > > >
> > > > > Finally, [cite 3] demonstrated that the Shapley interaction index $I^{\rm Shapley}(T|\mathbf{x})$ [cite 4] can be represented as the Harsanyi dividend $I_{\text{\rm and}}(S|\mathbf{x})$, i.e., $I^{\rm Shapley}(T|\mathbf{x})=\sum_{S\subseteq N \setminus T}\frac{1}{|S|+1} I_{\text{\rm and}}(S\cup T|\mathbf{x})$.
> > > > >
> > > > >
> > > > > [cite 1] Ren, J., et al. Defining and quantifying the emergence of sparse concepts in dnns. In CVPR, 2023.
> > > > >
> > > > > [cite 2] John C Harsanyi. A simplified bargaining model for the n-person cooperative game. International Economic Review, 1963.
> > > > >
> > > > > [cite 3] Li, M., et al. Does a neural network really encode symbolic concept? In ICML, 2023.
> > > > >
> > > > > [cite 4] Grabisch, M. and Roubens, M.. An axiomatic approach to the concept of interaction among players in cooperative games. International Journal of Game Theory,1999.

---

> ### Comment · Reviewer_f9us · 2023-11-23
>
> Thank you for the detailed example.
>
> Providing such an example in a concrete case brings the paper in right direction because it makes the paper more accessible and more likely to be reproducible. However, my concerns about the clarity of the main paper remain.  It would be helpful if the main paper included a clearer statement of both the method (more succinctly than your new appendix) and, especially if the main paper provided more of a discussion situating the new approach within the context of a broader range of related work.
>
> For example: once the masking approach of the paper is clarified, the paper should also clarify how it differs from, or how it addresses limitations of previous methods that use masking to explain the behavior of a network (these include Petsiuk 2019, Fong 2017, Dabkowski 2017, Hanjie Chen 2020,  and recently Johannes Linder 2022).  What problems with those previous research works are addressed by this new paper's methods?  The paper would benefit from a much more detailed discussion of other papers that have taken a related approach, and how they relate to the current work.
>
> If rewritten with both (1) this clarity of the current method and (2) and its relation to the broader research context, the paper would be stronger, but my recommendation is that the paper still needs a major revision to reach this level of clarity.  It would best be done as a major revision for another venue. (Remaining a 5)

---

> ### Author Response · Authors · 2023-11-23
>
> Thank you for your feedback. We are glad to answer your questions. However, if you still have new concerns, we have no time to answer your new feedback before the deadline.
>
> Compared to traditional masking methods, our approach is quite different.  Instead, our approach aims to obtain an essential representation of the model's inference logic. We try to mimic the final output of the neural network with a sparse, small number of interactions. In contrast, these traditional masking methods have fully different tasks and different motivations.

---

### Author Response · Authors · 2023-11-18

## Response to all reviewers

We would like to thank all reviewers for the constructive comments and questions. We have carefully considered all your comments and answered all the questions. In addition, we have conducted new experiments as requested. We hope that our responses would address your concerns.

We find that most concerns focus on the paper writing and the clarity of some technical details (e.g., detailed explanations of the meaning of an equation, the choice of $\alpha$, the running time, etc.). Besides, we have carefully revised the paper according to your suggestions. Please see the responses to each reviewer later.

**Please let us know as soon as possible, if you have any further questions or concerns.**

---

### Author Response · Authors · 2023-11-22

Thank you for all of your constructive comments and suggestions. Please let us know as soon as possible if you have any further questions or concerns.

---

### Meta-Review · Area_Chair_Qqy3 · 2023-12-16

**Metareview:**

This paper proposes a new approach for interpreting neural networks based on AND/OR interactions between a (small) set of input variables. To resolve the ambiguity in interpretation across different neural networks trained for the same task, the authors propose a method to extract interactions that are shared by these DNNs. The authors provide empirical evidence with BERT, Opt, LLama models to showcase the effectiveness of their approach. The reviewers universally acknowledged the importance of the problem and the simplicity of the approach. There were some concerns raised regarding the large variations in empirical performance across models, the clarity of the approach, and the correctness of the theoretical results. The authors were able to address these concerns, and incorporating them in the main paper will greatly improve the readability of the work.

**Justification For Why Not Higher Score:**

Clarity on several accounts was severely lacking. Somewhat unclear if this approach will be of significance to large-scale models due to the large variations in empirical performance.

**Justification For Why Not Lower Score:**

Only pending concern for one of the reviewers was clarity, which was addressed by the authors in the review and can be easily incorporated in detail for the camera-ready.

---

### Decision · Program_Chairs · 2024-01-16

Accept (poster)